# Vlaser: Vision-Language-Action Model with Synergistic Embodied Reasoning

**Ganlin Yang**[1,2*], **Tianyi Zhang**[4,2*], **Haoran Hao**[5,2*], **Weiyun Wang**[6,2], **Yibin Liu**[9,3], **Dehui Wang**[3]
**Guanzhou Chen**[3,2], **Zijian Cai**[10,3], **Junting Chen**[8,2], **Weijie Su**[2], **Wengang Zhou**[1], **Yu Qiao**[2]
**Jifeng Dai**[7,2], **Jiangmiao Pang**[2], **Gen Luo**[2], **Wenhai Wang**[2], **Yao Mu**[3,2†], **Zhi Hou**[2†]

[1]University of Science and Technology of China    [2]Shanghai AI Laboratory
[3]Shanghai Jiao Tong University  [4]Zhejiang University  [5]Nanjing University  [6]Fudan University
[7]Tsinghua University   [8]NUS   [9]Northeastern University   [10]Shenzhen University

Project Page: Vlaser

## Abstract

While significant research has focused on developing embodied reasoning capabilities using Vision-Language Models (VLMs) or integrating advanced VLMs into Vision-Language-Action (VLA) models for end-to-end robot control, few studies directly address the critical gap between upstream VLM-based reasoning and downstream VLA policy learning. In this work, we take an initial step toward bridging embodied reasoning with VLA policy learning by introducing **Vlaser** – a **V**ision-**L**anguage-**A**ction Model with **s**ynergistic **e**mbodied **r**easoning capability, which is a foundational vision-language model designed to integrate high-level reasoning with low-level control for embodied agents. Built upon the high-quality Vlaser-6M dataset, Vlaser achieves state-of-the-art performance across a range of embodied reasoning benchmarks—including spatial reasoning, embodied grounding, embodied QA, and task planning. Furthermore, we systematically examine how different VLM initializations affect supervised VLA fine-tuning, offering novel insights into mitigating the domain shift between internet-scale pre-training data and embodied-specific policy learning data. Based on these insights, our approach achieves state-of-the-art results on the WidowX benchmark and competitive performance on the Google Robot benchmark. The code, model and data are available at `https://github.com/OpenGVLab/Vlaser/`.

## 1 Introduction

Embodied artificial intelligence (AI) (Chrisley, 2003) aims to endow agents with the ability to perceive, understand, and act in the physical world. Achieving such intelligence requires not only accurate perception and language understanding but also embodied reasoning and effective control, which together define the paradigm of vision-language-action (VLA) models. Developing foundation models that possess strong reasoning and control capabilities is therefore an important advancement toward general-purpose embodied AI.

In this context, vision-language models (VLMs) (OpenAI, 2023; Liu et al., 2023; Chen et al., 2024; Bai et al., 2025; Team et al., 2023) emerge as natural candidates to enhance embodied agents in perception generalization and reasoning ability. Following this paradigm, extensive embodied vision-language models (Azzolini et al., 2025; Team et al., 2025c) emerge from enhancing the key ability for an embodied agent in grounding, planning, and spatial reasoning. Meanwhile, a significant body of work extends vision-language models (VLMs) into vision-language-action models (VLAs) (Kim et al., 2024; Intelligence et al., 2025; Driess et al., 2025) for robot control. While there are some approaches (Intelligence et al., 2025; Driess et al., 2025) that demonstrate the effectiveness of co-training with web data for the generalization in robot manipulation, it remains poorly understood which multi-modal data streams/abilities are most critical for improving downstream VLA models. In this paper, we aim to construct Vlaser, an embodied vision-language model that possesses strong

---

*Equal contribution. †Corresponding authors.

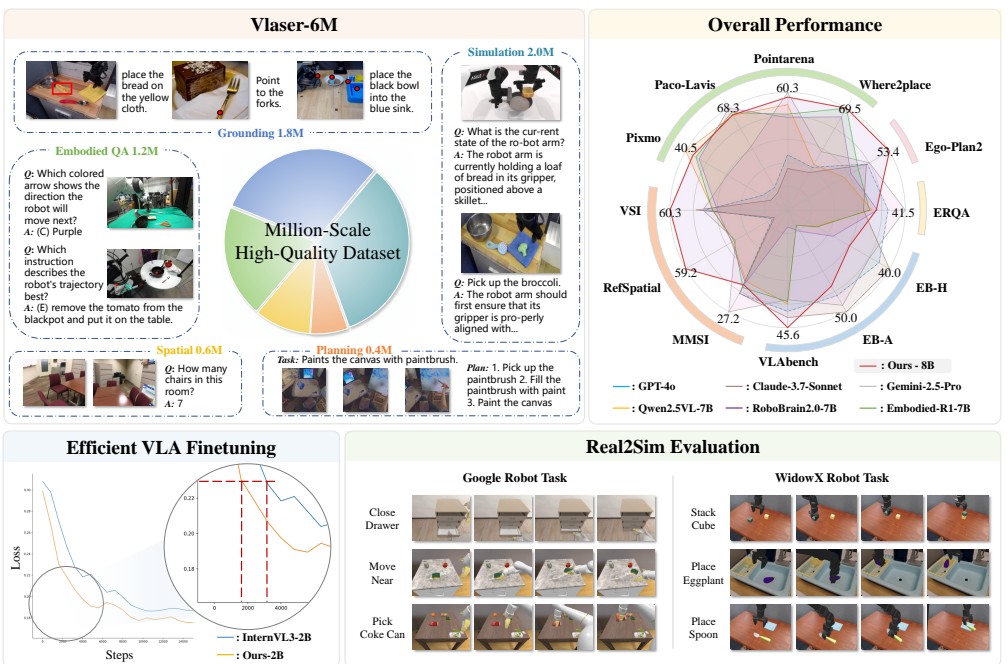

Figure 1: **Overall framework, capabilities, and evaluation of Vlaser. Top-left:** Composition of the Vlaser-6M dataset, featuring multi-task embodied data—including QA, grounding, spatial reasoning, and planning—along with in-domain simulation-sourced pairs. **Top-right:** A LiDAR visualization illustrating the state-of-the-art embodied reasoning capability of the Vlaser VLM. **Bottom-left:** The pre-trained Vlaser VLM significantly accelerates convergence in downstream Vision-Language Action model (VLA) policy learning on WidowX platform (Walke et al., 2023a). **Bottom-right:** Successful closed-loop operation of an agent powered by Vlaser within the SimplerEnv benchmark (Li et al., 2024c).

embodied reasoning capabilities, and subsequently answer this question based on the corresponding vision-language-action models.

Despite advancements in vision-language models (Chen et al., 2024; Bai et al., 2025), the capabilities of operating as an embodied agent remain severely constrained. In particular, navigation and traditional manipulation approaches rely heavily on planning-based control (Huang et al., 2022; Gasparetto et al., 2015; Zhang et al., 2018), which requires a strong foundational ability in grounding and planning. Planning and Grounding are cornerstones of the agents embodied in the physical world. Meanwhile, spatial understanding increasingly attracts the interest of the community in addressing the spatial perception ability of VLM. To this end, we firstly aim to introduce an embodied vision-language model specifically enhanced for the aboved embodied reasoning capabilities. Specifically, we construct the Vlaser data engine, which enables the systematic construction of the Vlaser-6M dataset by curating, reorganizing, and annotating public datasets from the Internet. As illustrated in Figure 1, the resulting dataset spans a wide spectrum of embodied reasoning tasks–including general embodied QA, visual grounding, spatial intelligence, task planning, and in-domain simulation data. Leveraging this comprehensive data foundation, Vlaser achieves state-of-the-art performance across a variety of embodied reasoning benchmarks, demonstrating strong generalization in both open-loop inference and closed-loop control settings.

Existing Vision-Language-Action (VLA) models (Black et al., 2024; Cheng et al., 2024; Kim et al., 2024; Intelligence et al., 2025) typically fine-tune pre-trained Vision-Language Models (VLMs) for robot control. However, the selection of an optimal VLM backbone – one that accelerates convergence and improves success rates when used as initialization for end-to-end VLA policy learning, remains an under-explored research problem. To address this gap, we systematically investigate the VLM-to-VLA adaptation paradigm using our enhanced embodied vision-language model and associated data engine. Our experiments reveal an important insight: although out-of-domain embodied reasoning

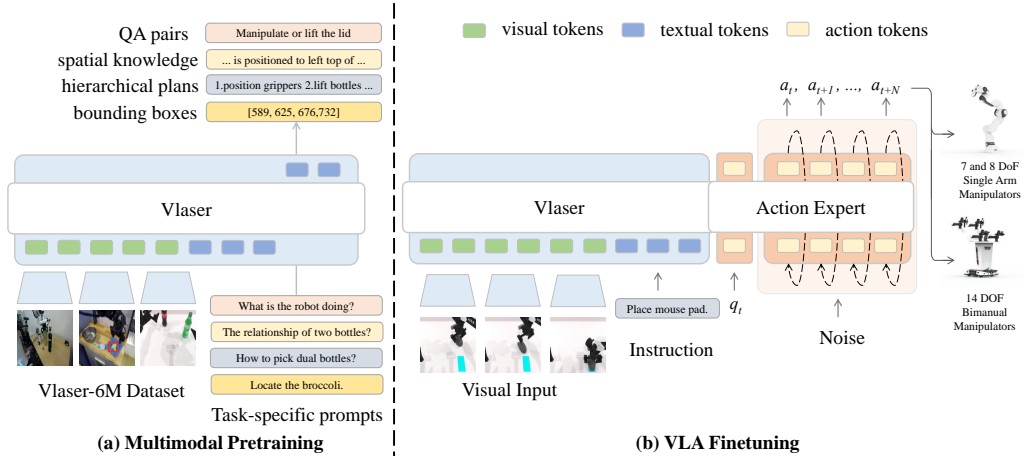

Figure 2: **An illustration of Vlaser architecture.** Vlaser includes two components and corresponding training phases: 1) the Multimodal Pretraining is for embodied reasoning enhancement based on the corresponding data engine; 2) VLA training is performed on the action expert module, which handles low-level control based on flow matching action generation.

data significantly improve upstream reasoning capabilities as measured by standard benchmarks, these gains may not translate directly or prominently to downstream VLA performance. In contrast, in-domain data – annotated directly on robot interaction datasets such as Open X-Embodiment (O'Neill et al., 2024) proves substantially more effective in accelerating convergence and increasing task success rates during VLA fine-tuning. We believe this observation provides significant insights for future embodied vision-language model construction: It is urgent to shrink the domain gap between current embodied perception and reasoning benchmarks to the real-world embodied tasks, and thus facilitate the closed-loop evaluation for the corresponding robot embodiment.

In summary, the principal contributions of Vlaser are as follows.

**An open-source embodied vision-language model and dataset.** We introduce Vlaser, an adaptable vision-language model that enhances InternVL with embodied reasoning capabilities and end-to-end robot control. The full model weights, modular data generation pipeline, training and evaluation code, and the accompanying Vlaser-6M dataset will be made publicly available to support reproducibility and future research.

**Systematic analysis of data effectiveness for VLA transfer.** We conduct a thorough investigation into which types of vision-language pretraining data contribute most effectively to downstream Vision-Language-Action (VLA) policy learning. Our findings offer practical insights for constructing task-aware data streams that bridge the gap between Internet-scale pretraining and embodied-specific fine-tuning.

**State-of-the-art performance across embodied benchmarks.** Among models of comparable scale, Vlaser achieves top-tier results on a comprehensive set of embodied reasoning benchmarks—spanning visual grounding, task planning, spatial reasoning, and simulation-based robot evaluation, demonstrating its strong generalization and applicability to both open-loop inference and closed-loop control scenarios.

## 2 METHOD

Vlaser aims to integrate embodied reasoning with end-to-end robot control for embodied agents, and identify the most crucial VLM data stream for VLA models. We first present the Vlaser structures in Section 2.1. Then, we illustrate the data engine in Section 2.2. Section 2.3 discusses the training recipe that includes embodied reasoning pretraining and vision-language-action finetuning.

## 2.1 MODEL STRUCTURE

The structure of Vlaser consists of two major components: the typical vision-language backbone (Chen et al., 2024; Liu et al., 2023) and the action expert for low-level control, as shown in Figure 2. We illustrate the two components respectively in this section.

**VLM Backbone** Vision-language models (VLMs) are key candidates for embodied agents, providing both perception and reasoning abilities. Vlaser , built on InternVL3 (Zhu et al., 2025), integrates embodied reasoning with robot control for embodied agents. While InternVL3 excels in multimodal and linguistic tasks across various model sizes, Vlaser focuses on two sizes—2B and 8B—optimized for the computational constraints of robots. These models utilize InternViT (Chen et al., 2024) as the vision encoder, paired with Qwen2.5-1.5B and Qwen2.5-7B LLMs (Qwen et al., 2025). Unlike typical multimodal MLLMs, Vlaser emphasizes embodied common-sense reasoning and end-to-end robot control capabilities.

**Action Expert** There are a large number of MLLMs (Team et al., 2025a; NVIDIA et al., 2025a) that enhance the ability of embodied common-sense reasoning for agents, while a few approaches equip the embodied MLLMs with end-to-end robot control. Vlaser extends the MLLMs with a low-level robot control and verifies the capability of different data streams in downstream VLA finetuning. Following (Intelligence et al., 2025), we design an action expert based on the open-source vision-language model (Chen et al., 2024; Zhu et al., 2025). Meanwhile, we utilize the flow matching (Lipman et al., 2023a) for action prediction based on the llava-like vision-language structure, while sharing the self-attention among the language model and action expert module. Specifically, we encode the robot state as a state token and noised actions as action tokens, and input them into the action expert. Meanwhile, we utilize non-causal attention for the VLA stream. During inference, we denoise the actions based on the image observation, language instruction, as well as the current robot state.

## 2.2 VLASER DATA ENGINE

This section outlines the composition of the Vlaser-6M data engine, a cornerstone for the model's embodied reasoning capabilities. Here we present the overall data scale and sources for each reasoning modality, while more details about the construction methodologies are provided in Appendix A.2.

**Embodied Grounding Data** The Vlaser dataset incorporates two distinct 2D grounding formats—bounding boxes and center points—both normalized to the range [0, 1000] to ensure consistent and resolution-invariant grounding predictions across diverse image resolutions. Specifically, we collect 1.5 million high-quality question-answer pairs that support multiple grounding tasks: predicting bounding boxes from open-vocabulary descriptions, localizing object center points based on textual descriptions, and identifying objects from given spatial coordinates. The data is sourced from several open embodied grounding datasets, including RoboPoint (Yuan et al., 2024), ShareRobot (Ji et al., 2025), Pixmo-Points (Deitke et al., 2025), Paco-LaVIS (Ramanathan et al., 2023), and RefSpatial (Zhou et al., 2025a). To further enhance generalization capabilities for open-world and open-vocabulary scenarios, we also generate an additional 300k point and bounding box annotations derived from segmentation masks in the SA-1B dataset (Kirillov et al., 2023). This combination of curated human annotations and synthetically enriched data aims to bolster both the diversity and scalability of visual grounding under real-world embodied settings.

**General and Spatial Reasoning Data** The Vlaser dataset integrates 1.2 million question-answer pairs dedicated to general Robotic Visual Question Answering (RoboVQA) tasks, along with an additional 500k data items specifically designed to enhance spatial intelligence. This comprehensive data composition substantially strengthens the model's capabilities in general state perception and 3D spatial reasoning. For the RoboVQA component, data is aggregated from multiple established sources, including RoboVQA (Sermanet et al., 2024), Robo2VLM (Chen et al., 2025b), RoboPoint (Yuan et al., 2024), RefSpatial (Zhou et al., 2025a), OWMM-Agent (Chen et al., 2025a), among others. To support spatial understanding and reasoning, we incorporate open-source datasets such as SPAR (Zhang et al., 2025), SpaceR-151k (Ouyang et al., 2025), and VILASR (Wu et al., 2025). Furthermore, we augment these with 100k manually annotated spatial understanding samples generated from publicly available 3D scene datasets—including ScanNet (Dai et al., 2017), ScanNet++ (Yeshwanth et al.,

2023), CA-1M (Lazarow et al., 2025), and ARKitScenes (Baruch et al., 2021). The integration of these diverse and high-quality data sources effectively enhances the model's spatial awareness and supports more robust performance in complex embodied reasoning tasks.

**Planning Data** To tackle complex tasks, it is essential to decompose them into manageable sub-tasks and solve them step by step. This capability is commonly referred to as planning. Effective planning allows robots to combine basic skills and generalize to new scenarios. We collected 400k training data to strengthen the model's planning ability, encompassing both language-based planning data and multimodal tasks. These include Alpaca-15k-Instruction (Wu et al., 2023) and MuEP (Li et al., 2024a). To further enhance environmental understanding and reasoning for complex decision-making, we incorporated training data with detailed reasoning processes from WAP (Shi et al., 2025). To improve the model's ability to comprehend complex instructions and execute tasks, we followed the annotations of LLaRP (Szot et al., 2024) to initialize planning tasks in Habitat (Szot et al., 2021) and generate planning trajectories to accomplish these tasks. In addition, we integrated egocentric video datasets such as EgoPlan-IT (Chen et al., 2023) and EgoCOT (Mu et al., 2023), which closely align with the observational perspective of embodied agents and provide valuable planning examples.

**In-Domain Data for downstream VLAs** To facilitate the end-to-end policy learning for Vision-Language Action Models (VLAs), we further generate 2 million in-domain multimodal question-answer pairs tailored for VLM pretraining. These data are specifically designed to align with the embodied reasoning context and enhance the model's ability to perceive, reason, and plan in interactive environments. The in-domain data is sourced from simulation platforms SimplerEnv (Li et al., 2024d) and RoboTwin (Chen et al., 2025c). Within SimplerEnv, data is generated for two distinct robotic embodiments: the Google Robot (Brohan et al., 2023b;a; O'Neill et al., 2024) and the WidowX Robot (Walke et al., 2023a), and within RoboTwin, data is generated from dual-arm Aloha-AgileX Robot. The question-answer pairs encompass the specialized categories including embodied grounding, spatial intelligence, planning and general VQA for robot states as described above. The detailed methodology for constructing and filtering each of the in-domain data in simulation is described in Appendix A.2.

## 2.3 TRAINING RECIPE

Vlaser adopts a two-stage training recipe, designed to optimize both embodied reasoning and robot control. It includes a VLM pretraining followed by a VLA finetuning. In this section, we elaborate on the training recipe among all phrases.

**Vision-Language Pretraining** Vlaser is developed by supervised fine-tuning (SFT) InternVL3 (Zhu et al., 2025) on embodied-related datasets, including those focused on grounding, planning, and spatial intelligence. In the first training phase, we fine-tune InternVL3 using auto-regressive language modeling loss. In particular, given the input images $x \in \mathbb{R}^{t \times h \times w \times 3}$ and textual prompt $y \in \mathbb{R}^l$, the language modeling loss $\mathcal{L}_{lm}$ can be defined by

$$\mathcal{L}_{lm} = -\log p(t_N | \mathcal{F}_\text{v}(x; \theta_v), \mathcal{F}_\text{t}(y), t_{0:N-1}; \Theta), \tag{1}$$

where $p \in \mathbb{R}^m$ is the next-token probability and $m$ denotes the vocabulary size. Here, $\mathcal{F}_\text{v}(\cdot)$ denotes the ViT and the MLP, and $\theta_v$ is their parameters. $\mathcal{F}_\text{t}(\cdot)$ is the textual tokenizer. $\Theta$ are the parameters of the LLM. $t_i$ denotes the *i-th* predicted word.

**Vision-Language-Action Finetuning** For robot policy learning, we optimize the model using an additional incorporated action expert module trained on robot-specific datasets. The action expert is analogous to a mixture of experts(MoE) (Shazeer et al., 2017b; Du et al., 2022; Zhou et al., 2024) architecture with two mixture elements, while the original part of parameter is used for image and text inputs, and the additionally separate set of weights for the robotics-specific (action and state) tokens inputs and outputs are referred as the *action expert*. Vlaser integrates a flow-matching-based action expert to predict a sequence of future actions from a single-frame observation. Specifically, we denote the action chunk $\mathbf{A}_t = [\mathbf{a}_t, \mathbf{a}_{t+1}, \ldots, \mathbf{a}_{t+H-1}]$, where $\mathbf{a}_t$ represents the action in the current timestep $t$ and $H$ represents the action horizon. $\mathbf{o}_t = [\mathbf{I}_t^1, ..., \mathbf{I}_t^n, l_t, \mathbf{q}_t]$ indicates the observations (image $\mathbf{I}_t^i$ with $n$ views, language $l_t$ and robot state $\mathbf{q}_t$) at action timestep $t$. $\mathbf{I}_t^i$, $l_t$ and $\mathbf{q}_t$ are encoded via corresponding encoders and then projected via a linear projection layer into the same embedding space. $\theta$ represents the action expert network and $\tau \in [0, 1]$ represents the flow matching timesteps.

Table 1: **Comparison with existing close-sourced, open-sourced and embodied-related VLMs on 12 general embodied reasoning benchmarks, spanning from embodied QA, planning, embodied grounding to spatial intelligence and close-loop simulation evaluation.** Avg denotes the normalized average performance of all the benchmarks. The best, second best and third best score among all the baselines are colored in red, orange and yellow.

| Model | QA | Planning | Embodied Grounding | | | | Spatial Intelligence | | | Simulation | | | Avg |
|---|---|---|---|---|---|---|---|---|---|---|---|---|---|
| | ERQA | Ego-Plan2 | Where2place | Pointarena | Paco-Lavis | Pixmo-Points | VSIBench | RefSpatial | MMSIBench | VLABench | EB-ALFRED | EB-Habitat | |
| ▼ *Closed-source MLLMs:* | | | | | | | | | | | | | |
| GPT-4o-20241120 | 47.0 | 41.8 | 29.1 | 29.5 | 16.2 | 10.8 | 42.5 | 8.8 | 30.3 | 39.3 | 56.3 | 59.0 | 34.2 |
| Claude-3.7-Sonnet | 35.5 | 41.3 | 25.6 | 22.2 | 12.4 | 7.2 | 47.0 | 7.7 | 30.2 | 41.7 | 67.0 | 65.7 | 33.6 |
| Gemini-2.5-Pro | 55.0 | 42.9 | 39.9 | 62.8 | 45.5 | 25.8 | 43.4 | 30.3 | 36.9 | 34.8 | 62.7 | 53.0 | 44.4 |
| ▼ *Small Size MLLMs:* | | | | | | | | | | | | | |
| ChatVLA-2B | 34.3 | 25.3 | 3.7 | 10.1 | 10.2 | 2.1 | 2.4 | 0.9 | 20.1 | 0.0 | 0.0 | 0.0 | 9.1 |
| InternVL3-2B | 31.5 | 30.9 | 5.2 | 7.1 | 15.4 | 1.4 | 31.5 | 1.8 | 25.3 | 19.4 | 1.3 | 12.0 | 15.2 |
| Qwen2.5VL-3B | 35.3 | 30.3 | 31.0 | 41.7 | 67.4 | 36.6 | 27.9 | 24.9 | 26.5 | 31.3 | 6.7 | 19.7 | 31.6 |
| Embodied-R1-3B | 36.0 | 36.0 | 35.1 | 45.3 | 68.3 | 36.6 | 28.0 | 28.5 | 26.0 | 24.6 | 7.0 | 19.3 | 32.5 |
| RoboBrain2.0-3B | 37.3 | 41.8 | 64.2 | 46.0 | 67.6 | 36.9 | 28.8 | 46.5 | 26.8 | 18.1 | 0.0 | 10.0 | 35.3 |
| Vlaser-2B | 35.8 | 38.3 | 74.0 | 57.8 | 72.5 | 44.6 | 57.5 | 43.0 | 23.6 | 23.1 | 42.3 | 30.7 | 45.3 |
| ▼ *Medium Size MLLMs:* | | | | | | | | | | | | | |
| Magma-8B | 29.3 | 27.9 | 10.9 | 29.6 | 15.3 | 10.1 | 12.7 | 4.5 | 26.2 | 8.5 | 0.0 | 0.0 | 14.6 |
| Cosmos-Reason1-7B | 39.3 | 26.9 | 11.4 | 40.8 | 61.8 | 23.6 | 33.9 | 5.4 | 26.4 | 35.5 | 4.0 | 5.3 | 26.2 |
| VeBrain-7B | 38.3 | 27.3 | 33.1 | 38.9 | 55.1 | 20.1 | 39.9 | 20.6 | 28.3 | 25.9 | 5.7 | 12.3 | 28.8 |
| InternVL3-8B | 35.3 | 40.0 | 10.0 | 14.2 | 21.1 | 5.7 | 42.1 | 5.6 | 25.7 | 24.7 | 19.0 | 23.7 | 22.3 |
| Qwen2.5VL-7B | 39.3 | 29.7 | 31.1 | 56.3 | 68.0 | 43.5 | 38.2 | 32.1 | 25.9 | 36.4 | 10.0 | 18.3 | 35.7 |
| Embodied-R1-7B | 38.3 | 37.1 | 69.5 | 51.2 | 69.9 | 39.2 | 38.6 | 31.1 | 28.1 | 35.5 | 10.0 | 19.0 | 38.9 |
| RoboBrain2.0-7B | 42.0 | 33.2 | 63.6 | 49.5 | 73.1 | 37.8 | 36.1 | 32.5 | 26.5 | 6.6 | 14.0 | 29.3 | 37.0 |
| Vlaser-8B | 41.0 | 53.4 | 69.5 | 60.3 | 68.3 | 40.5 | 60.3 | 59.2 | 27.2 | 45.6 | 50.0 | 40.0 | 51.3 |

For the action chunk, $\mathbf{A}_t^\tau = \tau \mathbf{A}_t + (1 - \tau)\epsilon$ is the corresponding noisy action chunk, and we train the network to output $\mathbf{v}_\theta(\mathbf{A}_t^\tau, \mathbf{o}_t)$ to match the denoising vector field $\mathbf{u}(\mathbf{A}_t^\tau | \mathbf{A}_t) = \epsilon - \mathbf{A}_t$ where $\epsilon \sim \mathcal{N}(\mathbf{0}, \mathbf{I})$. Therefore, the VLA optimization loss is as follows,

$$\mathcal{L}_{vla} = \mathbb{E}_{p(\mathbf{A}_t | \mathbf{o}_t)} \left\| \mathbf{v}_\theta(A_t^\tau, \mathbf{o}_t) - \mathbf{u}(\mathbf{A}_t^\tau | \mathbf{A}_t) \right\|^2 \tag{2}$$

Formally, following prior flow-matching based VLA works (Black et al., 2024; Zren, 2025), We sample the action chunks from the robot episodes and flow-matching timesteps to optimize the network. At inference, we generate actions by integrating the learned vector field from $\tau = 0$ to $\tau = 1$, starting with random noise $\mathbf{A}_t^0 \sim \mathcal{N}(\mathbf{0}, \mathbf{I})$, as follows,

$$\mathbf{A}_t^{\tau+\delta} = \mathbf{A}_t^\tau + \delta \mathbf{v}_\theta(\mathbf{A}_t^\tau, \mathbf{o}_t) \tag{3}$$

where $\delta$ is the integration step size. In our experiments, we set $H$ as 4, and $\delta$ as $0.1(\delta^{-1} = 10$ integration steps) at inference time for the improvement of inference efficiency. We aim to identify the most effective VLMs for downstream VLA fine-tuning and bridge the gap between foundational VLMs and their performance in downstream VLA tasks, thus shedding light on the future construction of embodied VLMs. Currently, the SimplerEnv benchmark, including Bridge (Walke et al., 2023b) and Google Robot (Jang et al., 2022; Brohan et al., 2023b) datasets, provides numerous training episodes (Over 5M images) and corresponding Real-to-Sim benchmarks. We thus majorly analyze the most effective data stream for VLA finetuning based on SimplerEnv.

## 3 EXPERIMENTS

### 3.1 PERFORMANCE ON EMBODIED REASONING CAPABILITY

**Evaluation Datasets** We conduct a comprehensive evaluation of embodied reasoning capabilities across a total of 12 benchmarks, covering a wide spectrum of tasks including embodied question answering, task planning, embodied grounding, spatial intelligence, and closed-loop simulation evaluation. The evaluated benchmarks consist of: ERQA (Team et al., 2025b), Ego-Plan2 (Qiu et al., 2024), Where2place (Yuan et al., 2024), Pointarena (Cheng et al., 2025), Paco-Lavis (Ramanathan et al., 2023), Pixmo-Points (Deitke et al., 2025), VSI-Bench (Yang et al., 2025b), RefSpatial-Bench (Zhou et al., 2025a), MMSI-Bench (Yang et al., 2025d), VLABench (Zhang et al., 2024), and EmbodiedBench (Yang et al., 2025c). For EmbodiedBench, we further assess performance in two simulation environments ALFRED (Shridhar et al., 2020) and Habitat (Szot et al., 2021).

Table 2: **SimplerEnv Evaluation on WidowX Robot Tasks.** Avg indicates the average success rate among the four tasks. Model sizes are indicated within parentheses. The result of RT-1-X (Brohan et al., 2023b), Octo-Base (Team et al., 2024), OpenVLA (Kim et al., 2024), RoboVLM (Liu et al., 2025) and SpatialVLA (Qu et al., 2025b) are from (Qu et al., 2025b) while the results of $\pi_0$ (Black et al., 2024) is from (Zren, 2025).

| Model | Carrot on plate | Put eggplant in basket | Spoon on towel | Stack Cube | Avg |
|---|---|---|---|---|---|
| RT-1-X (35M) (Brohan et al., 2023b) | 4.2% | 0% | 0% | 0% | 1.1% |
| Octo-Base (93M) (Team et al., 2024) | 8.3% | 43.1% | 12.5% | 31.9% | 16.0% |
| OpenVLA (7B) (OpenAI, 2023) | 0% | 4.1% | 0% | 0% | 1.0% |
| RoboVLM (2B) (Liu et al., 2025) | 25.0% | 58.3% | 29.2% | 12.5% | 31.3% |
| SpatialVLA (4B) (Qu et al., 2025b) | 25.0% | 100.0% | 16.7% | 62.5% | 42.7% |
| $\pi_0$ (3B) (Black et al., 2024) | 55.8% | 79.2% | 63.3% | 21.3% | 54.9% |
| InternVL3-2B | 42.9% | 57.1% | 55.8% | 11.3% | 41.8% |
| Vlaser-OOD (2B) | 60.8% | 35.4% | 56.7% | 20.0% | 43.2% |
| Vlaser-QA (2B) | 55.8% | 83.3% | 77.9% | 33.3% | 62.6% |
| Vlaser-Spatial (2B) | 48.3% | 81.7% | 76.7% | 36.7% | 60.8% |
| Vlaser-Grounding (2B) | 47.5% | 80.8% | 80.0% | 39.6% | 62.0% |
| Vlaser-All (2B) | 52.5% | 87.9% | 76.6% | 43.3% | 65.1% |

**Baselines** Since our method, Vlaser is trained at two model scales – 2B and 8B parameters, we categorize the compared baseline methods into three groups for a systematic evaluation: 1) **State-of-the-art closed-source models**, including GPT-4o (OpenAI, 2025), Claude-3.7-Sonnet (Anthropic, 2025), and Gemini-2.5-Pro (Comanici et al., 2025); 2) **Small-scale MLLMs (2B – 3B parameters)**, comprising ChatVLA-2B (Zhou et al., 2025b), InternVL3-2B (Zhu et al., 2025), Qwen2.5-VL-3B (Bai et al., 2025), Embodied-R1-3B (Yuan et al., 2025), and RoboBrain2.0-3B (Team et al., 2025a); 3) **Medium-scale MLLMs (7B – 8B parameters)**, including Magma-8B (Yang et al., 2025a), Cosmos-Reason1-7B (NVIDIA et al., 2025a), VeBrain-7B (Luo et al., 2025), InternVL3-8B (Zhu et al., 2025), Qwen2.5-VL-7B (Bai et al., 2025), Embodied-R1-7B (Yuan et al., 2025), and RoboBrain2.0-7B (Team et al., 2025a).

The overall experimental results are presented in Table 1. As shown in Table 1, compared to the base models InternVL3-2B and InternVL3-8B used as initialization for our supervised finetuning, our Vlaser yields substantial improvements across all embodied reasoning capabilities, with particularly notable gains in embodied grounding and simulation-based evaluation. For example, the average score increases from 15.2 to 45.3 for the 2B model, and from 22.3 to 51.3 for the 8B model. *These significant performance gains underscore the high quality and effectiveness of the Vlaser-6M dataset in enhancing embodied reasoning abilities.* An interesting observation emerges that when finetuning on the same Vlaser-6M dataset, a smaller sized Vlaser-2B outperforms Vlaser-8B on simple point grounding tasks that require direct, short answers. Conversely, Vlaser-8B demonstrates superior performance on more complex tasks such as multi-step planning and closed-loop simulation evaluation, which often benefit from chain-of-thought (CoT) reasoning. This scaling behavior indicates the importance of appropriate model size selection based on target application requirements.

When compared against current state-of-the-art embodied-specific VLMs, including Robo-Brain2.0 (Team et al., 2025a) and Embodied-R1 (Yuan et al., 2025), our method, Vlaser still achieves superior performance on the majority of benchmarks while remaining highly competitive on the remainder, **ultimately attaining the highest overall score** (by +10% margin overall). These results indicate that Vlaser delivers a well-balanced and robust capability set, performing strongly across multiple dimensions of embodied intelligence – from embodied question answering and state estimation to future action planning, visual grounding, spatial reasoning, and closed-loop simulation. Such comprehensive competence highlights its suitability as a versatile backbone for embodied AI brains. In the following section, we further examine how these enhanced reasoning capabilities, embedded within VLMs, translate into improved performance when fine-tuned for downstream Vision-Language Action models (VLAs) in simulation manipulation scenarios.

## 3.2 PERFORMANCE ON DOWNSTREAM CLOSE-LOOP ROBOT TASKS

**Finetuning Datasets** We firstly conduct extensive experiments on SimplerENV (Li et al., 2024d) to evaluate the performance of Vlaser and Vlaser data engine on closed-loop robotic manipulation

Table 3: **Comparison with existing methods in SimplerEnv on Google Robot tasks.** Avg indicates the average success rate among the three tasks. Model sizes are indicated within parentheses. The results of TraceVLA (Zheng et al., 2024), RT-1-X (Brohan et al., 2023b), Octo-Base (Team et al., 2024), OpenVLA (Kim et al., 2024), RoboVLM (Liu et al., 2025), Emma-X (Sun et al., 2024), Magma (Yang et al., 2025a), GR00T N1.5(NVIDIA et al., 2025b) and $\pi_0$ (Black et al., 2024) are from (Lee et al., 2025).

| Model | Visual Matching | | | Avg | Variant Aggregation | | | Avg |
|---|---|---|---|---|---|---|---|---|
| | Pick Coke Can | Move Near | Drawer | | Pick Coke Can | Move Near | Drawer | |
| TraceVLA (7B) (Zheng et al., 2024) | 28.0% | 53.7% | 57.0% | 42.0% | 60.0% | 56.4% | 31.0% | 45.0% |
| RT-1-X (35M) (Brohan et al., 2023b) | 56.7% | 31.7% | 59.7% | 53.4% | 49.0% | 32.3% | 29.4% | 39.6% |
| Octo-Base (93M) (Team et al., 2024) | 17.0% | 4.2% | 22.7% | 16.8% | 0.6% | 3.1% | 1.1% | 1.1% |
| OpenVLA (7B) (Kim et al., 2024) | 16.3% | 46.2% | 35.6% | 27.7% | 54.5% | 47.7% | 17.7% | 39.8% |
| RoboVLM (2B) (Liu et al., 2025) | 77.3% | 61.7% | 43.5% | 63.4% | 75.6% | 60.0% | 10.6% | 51.3% |
| Emma-X (7B) (Sun et al., 2024) | 2.3% | 3.3% | 18.3% | 8.0% | 5.3% | 7.3% | 20.5% | 11.0% |
| Magma (8B) (Yang et al., 2025a) | 56.0% | 65.4% | 83.7% | 68.4% | 53.4% | 65.7% | 68.8% | 62.6% |
| GR00T N1.5 (2.1B) (NVIDIA et al., 2025b) | 69.3% | 68.7% | 35.8% | 52.4% | 46.7% | 62.9% | 17.5% | 43.7% |
| $\pi_0$ (3B) (Black et al., 2024) | 72.7% | 65.3% | 38.3% | 58.3% | 75.2% | 63.7% | 25.6% | 54.8% |
| InternVL3-2B | 94.3% | 78.8% | 19.0% | 64.0% | 80.4% | 72.7% | 11.1% | 54.7% |
| Vlaser-OOD(2B) | 85.0% | 76.3% | 44.9% | 68.7% | 74.4% | 69.2% | 10.3% | 51.3% |
| Vlaser-QA (2B) | 90.0% | 84.2% | 44.4% | 72.9% | 78.2% | 78.2% | 13.0% | 56.4% |
| Vlaser-Spatial (2B) | 83.0% | 77.9% | 56.0% | 72.3% | 77.7% | 73.2% | 13.2% | 54.7% |
| Vlaser-Grounding (2B) | 83.3% | 83.3% | 54.2% | 73.6% | 81.2% | 76.8% | 17.0% | 58.3% |
| Vlaser-All (2B) | 91.0% | 85.4% | 52.1% | 76.2% | 80.5% | 77.7% | 18.8% | 59.0% |

tasks. SimplerENV is an open-source suite of purpose-built simulated environments with nearly 150K episodes for evaluating real-world robot manipulation policies in a scalable, reproducible way. It targets the key real-to-sim gaps – control and vision so that simulated performance reliably tracks real-robot outcomes. Across Google Robot and WidowX/BridgeData V2 setups, SimplerEnv reports strong real-vs-sim correlations and faithfully reflects behavior under distribution shifts, enabling fast, comparable policy assessment without full digital twins. As a result, SimplerENV has been widely adopted for evaluating VLA models and has proven to reliably reflect the performance of the models on the real robot platform. To further demonstrate the generalizability of our method to other simulation platforms and embodiments, we also conduct experiments on RoboTwin (Mu et al., 2024; Chen et al., 2025c) platforms with Aloha-AgileX as bimanual embodiement. Robotwin is a scalable framework for bimanual manipulation, which integrates scalable training sets and pre-defined tasks as benchmarks for comprehensive robust bimanual manipulation.

Table 4: **Robotwin Evaluation on Aloha-AgileX Robot Tasks.** Avg indicates the average success rate among the 12 tasks. The results of RDT-1B (Liu et al., 2024) are from our self-implemented training for 30k steps, which aligns the training setting with Vlaser.

| Simulation Task | RDT-1B (Liu et al., 2024) | InternVL3-2B | Vlaser-OOD(2B) | Vlaser-QA (2B) | Vlaser-Spatial (2B) | Vlaser-Grounding (2B) | Vlaser-All (2B) |
|---|---|---|---|---|---|---|---|
| Beat block hammer | 28% | 12% | 20% | 18% | 20% | 32% | 40% |
| Click bell | 46% | 78% | 94% | 48% | 98% | 86% | 92% |
| Handover mic | 92% | 74% | 56% | 84% | 60% | 52% | 84% |
| Move can pot | 44% | 40% | 42% | 66% | 56% | 50% | 46% |
| Move pillbottles pad | 10% | 62% | 70% | 66% | 78% | 68% | 72% |
| Move playingcard away | 20% | 64% | 52% | 58% | 68% | 84% | 74% |
| Pick diverse bottles | 2% | 30% | 44% | 36% | 24% | 34% | 38% |
| Place burger fries | 42% | 36% | 46% | 88% | 82% | 46% | 42% |
| Place container plate | 82% | 72% | 70% | 84% | 78% | 82% | 84% |
| Place phone stand | 8% | 42% | 40% | 38% | 36% | 48% | 50% |
| Place mouse pad | 2% | 68% | 30% | 44% | 38% | 48% | 92% |
| Shake bottle | 66% | 92% | 90% | 98% | 96% | 98% | 96% |
| Avg. | 36.8% | 55.8% | 54.5% | 60.7% | 61.2% | 60.7% | 67.5% |

**Baselines** Alongside comparisons with other commonly used VLA models (Black et al., 2024; Kim et al., 2024; Liu et al., 2025), we conduct a clear self-comparable ablation study to evaluate the individual contributions of different in-domain data sources. Specifically, **InternVL3-2B** denotes the base InternVL model, while **Vlaser-OOD** refers to the Vlaser-2B model fine-tuned solely on Vlaser-6M out-of-domain (OOD) data specific for the embodied reasoning benchmarks in Sec. 3.1, without any in-domain data in Vlaser-6M dataset. Regarding the in-domain datas derived directly from the simulation platform, we categorize them into three types: embodied QA (including planning), embodied spatial intelligence, and embodied grounding. The corresponding fine-tuned Vlaser-2B models

Table 5: **Ablation Studies on WidowX Robot Tasks**

| Model | Predict Length | Execute Length | Sample Steps | Carrot on plate | Put eggplant in basket | Spoon on the towel | Stack cube | Avg |
|---|---|---|---|---|---|---|---|---|
| InternVL-2B | 4 | 4 | 10 | 42.9% | 57.1% | 55.8% | 11.3% | 41.8% |
| | 4 | 2 | 10 | 22.9% | 18.3% | 40.8% | 2.9% | 21.2% |
| | 2 | 2 | 10 | 34.6% | 22.9% | 54.2% | 2.9% | 28.7% |
| | 4 | 4 | 20 | 38.8% | 54.2% | 51.3% | 8.3% | 38.2% |
| Vlaser-OOD(2B) | 4 | 4 | 10 | 60.8% | 35.4% | 56.7% | 20.0% | 43.2% |
| | 4 | 2 | 10 | 50.0% | 21.7% | 30.0% | 12.1% | 28.5% |
| | 2 | 2 | 10 | 62.5% | 19.2% | 49.2% | 21.3% | 38.1% |
| | 4 | 4 | 20 | 57.5% | 29.2% | 54.6% | 17.1% | 39.6% |
| Vlaser-QA(2B) | 4 | 4 | 10 | 55.8% | 83.3% | 77.9% | 33.3% | 62.6% |
| | 4 | 2 | 10 | 44.2% | 64.2% | 59.6% | 36.3% | 51.1% |
| | 2 | 2 | 10 | 47.5% | 66.3% | 67.1% | 36.3% | 54.3% |
| | 4 | 4 | 20 | 56.3% | 85.0% | 76.7% | 35.0% | 63.3% |

are denoted as **Vlaser-QA**, **Vlaser-Spatial**, and **Vlaser-Grounding**, respectively. Additionally, we combine all the three in-domain data types to fine-tune a model referred to as **Vlaser-All**.

The full experimental results are presented in Table 2, Table 3 and Table 4. Notably, no clear improvement was observed when using Vlaser-OOD-2B as the initial backbone across all three benchmarks. The task success rates of **Vlaser-OOD** and the baseline **InternVL3-2B** remain close. Conversely, all models fine-tuned with in-domain data—**Vlaser-QA**, **Vlaser-Spatial**, **Vlaser-Grounding**, and **Vlaser-All**—exhibit significant performance gains, even though the model architecture and size remain unchanged. This observation illustrates the effectiveness of our Vlaser data engine, and meanwhile identifies that *there is no positive correlation between common embodied reasoning benchmarks and the performance of closed-loop control of the lower level for the specific embodiment of the robot*. We reckon it is the domain shift between the internet data and the corresponding robot embodiment (e.g., WidowX or Google Robot), and we find that the enhanced abilities in the same observation domain effectively facilitate the closed-loop success rate. Therefore, it is urgent to shrink the domain gap between the foundational models and real-world robot embodiment for closed-loop task completion.

Regarding the three types of in-domain data annotations, we experimentally find that incorporating any of them leads to significant performance gains over the baseline. We attribute these improvements primarily to the mitigation of visual observation domain shift. Furthermore, integrating all three types of in-domain data results in further performance enhancement. *This suggests that pretraining with diverse in-domain multimodal data, spanning general QA, grounding, and spatial intelligence, could best facilitates transfer learning for VLA policy learning and leads to improved task success rates.*

## 3.3 ABLATION STUDIES

In this section, we adopt ablation studies regarding three key hyperparameters for VLA end-to-end training, *i.e.,*, the predicted action length $P$, the execute action length sizes $H$ as well as flow-matching sampling steps $\delta^{-1}$. By default we use $P$ as 4, with $H$ as 4, and $\delta^{-1}$ as 10. The results are shown in Table 5. It is clear to see that, no matter in which group of hyperparameter settings, the performance based on **Vlaser-OOD** shows slight improvement compared with the performance based on **InternVL3-2B**. Besides, there is a significant gains while using the model fine-tuned with in-domain data **Vlaser-QA**. This conclusion is as same as the results in 3.2, which demonstrates great robustness of our method.

## 4 RELATED WORK

**Vision-Language Model for Embodied Reasoning** Enhancing the embodied reasoning capabilities of current state-of-the-art Vision-Language Models (VLMs) has emerged as a critical research direction. These capabilities encompass a range of competencies, including grounding (Yuan et al., 2024; Deitke et al., 2025; Cheng et al., 2025) which identifies affordances that enable embodied agents to perform manipulations, spatial intelligence (Yang et al., 2025b;d), such as object counting and spatial relationship understanding, as well as task planning (Chen et al., 2023; Qiu et al., 2024), which involves assessing the current state and determining subsequent actions to be executed. Gemini Robotics-ER (Team et al., 2025b) integrates embodied reasoning into its core visual-language model (VLM), in parallel, a number of data-driven methodologies have emerged to support such reasoning

capabilities. For instance, Cosmos-Reason1 (NVIDIA et al., 2025a), VeBrain (Luo et al., 2025), MolmoAct (Lee et al., 2025), and EmbodiedOneVision (Qu et al., 2025a) each contribute curated datasets specifically designed for embodied reasoning tasks, emphasizing aspects such as multi-modal instruction following and action-aware visual-language alignment. Furthermore, several frameworks – including the RoboBrain series (Ji et al., 2025; Team et al., 2025a), Embodied R1 (Yuan et al., 2025), and Robix (Fang et al., 2025) incorporate Reinforcement Fine-Tuning (RFT) and synthesize spatio-temporal reasoning datasets enriched with structured thought traces. These approaches aim to enhance models' capacity for causal reasoning and long-horizon task decomposition. Distinguished from these prior efforts, our work not only achieves competitive, and in some cases superior performance on established embodied reasoning benchmarks, but also provides an in-depth analysis of the synergistic relationship between pre-trained VLMs and downstream Vision-Language Action Models (VLAs), offering insights that bridge model capabilities and real-world deployment.

**Vision-Language-Action models.** Developing a generalist model remains a central challenge in robotics. Inspired by the strong generalization abilities of vision-language models (VLMs) (OpenAI, 2023; Chen et al., 2024; Bai et al., 2025; Team et al., 2023) trained on large-scale internet data, researchers have proposed vision-language-action (VLA) models, which have demonstrated promising performance (Brohan et al., 2023a; Kim et al., 2024; Qu et al., 2025b; Hou et al., 2025). Compared to traditional robot policies, VLA models are pretrained on large-scale robotics datasets and exhibit improved generalization across object categories and visual observations. Building on recent progress, researchers have incorporated techniques such as diffusion (Ho et al., 2020; Rombach et al., 2022; Peebles & Xie, 2023), flow matching (Lipman et al., 2023b), and mixture-of-experts (MoE) (Shazeer et al., 2017a) into VLA models, and have adopted larger, more capable VLMs as their backbones. These advances have enabled VLA models to tackle a wider range of complex real-world manipulation tasks. Efforts have been made to enhance specific reasoning abilities in embodied scenarios (Team et al., 2025b; Ji et al., 2025; NVIDIA et al., 2025a). In parallel, several studies (Intelligence et al., 2025; Driess et al., 2025; Zhou et al., 2025b) explore unified training frameworks for VLMs and VLAs to leverage the reasoning capacity of VLMs. However, the relationship between high-level multimodal reasoning and low-level control performance remains largely unexplored. It is still unclear which specific multimodal abilities such as spatial understanding, grounding, or planning and which types of training data most effectively enhance the control capabilities of a VLA model. In this work, we take an initial step toward analyzing this relationship by a systematic evaluation, and also propose the latest foundational model with strong embodied multimodal understanding and action prediction.

## 5 LIMITATIONS

One limitation of this work is the absence of real-robot experiments. While the experiments conducted in this study have all been performed well in simulation, in future work we plan to conduct more experiments with real robots to further evaluate and refine the proposed methods.

## 6 CONCLUSION AND DISCUSSION

We introduce Vlaser, a foundational vision-language-action model that extends vision-language models with embodied reasoning and end-to-end robot control capabilities. Powered by the Vlaser-6M dataset, the model establishes a new state of the art across a wide range of embodied reasoning benchmarks, including planning, grounding, spatial reasoning, and simulation-based tasks. Moreover, Vlaser reveals the most effective data streams for downstream VLA through its curated data pipeline, achieving state-of-the-art performance on Bridge and competitive results on Google Robot for end-to-end robot control.

In this work, we reveal that current embodied reasoning benchmarks exhibit a significant domain gap when compared to real-world robots. This core domain shift arises from the observation that robots have a fundamentally different viewpoint from that of internet datasets. Additionally, there are inherent limitations due to the lack of sufficient data from the robot's perspective, despite the abundance of vision datasets available. Therefore, we argue that it is essential to develop alignment techniques to bridge the domain gap in representations between the robot's viewpoint and that of internet datasets.

## ACKNOWLEDGMENTS

This work is supported by Shanghai Artificial Intelligence Laboratory.

## ETHICS STATEMENT

This research adheres to the ICLR 2026 ethical guidelines and upholds the principles of responsible research. We ensure that no personally identifiable, sensitive, or harmful data were used. Our experiments were based on publicly available datasets and did not involve any human subjects or vulnerable groups. We have considered the potential societal impact of our methods, including the risk of misuse, and believe that these contributions primarily advance scientific understanding and do not pose foreseeable harm.

## REPRODUCIBILITY STATEMENT

We follow the reproducibility guidelines in the ICLR 2026 author guidelines. We will open source code, configuration files, and scripts to reproduce our results, including dataset construction, model training, and evaluation, on platforms such as GitHub and Huggingface as soon as possible.

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

# A APPENDIX

## THE USE OF LARGE LANGUAGE MODELS (LLMS)

We used a large language model, ChatGPT (OpenAI, 2025), only for grammar check and correction. All the writings was manually reviewed by the authors. Crucially, AI was not used to generate any research data, statistical analyses, figures, or conclusions. The authors take full responsibility for the final content and any errors herein.

## A.1 TRAINING DETAILS

Our Vlaser is optimized in a fully supervised finetuning (SFT) manner based on InternVL3 series (Zhu et al., 2025)(InternVL3-2B and InternVL3-8B). To maximize adaptation to embodied reasoning tasks, we keep all parameters trainable, including those in the large language model, the vision-language projector, and the visual encoder, enabling comprehensive end-to-end learning. Further details regarding the training setup, including hyperparameters and optimization settings, are provided in Table 6.

Table 6: **Hyper-parameters used in the VLM pretraining of Vlaser.**

| Configurations | Values |
| --- | --- |
| LLM sequence length | 16, 384 |
| Dynamic Resolution | True |
| Patch Size | 448 |
| Max Patch num | 12 |
| Freeze vision tower | False |
| Freeze multimodal projector | False |
| Freeze language model | False |
| Optimizer | AdamW |
| Optimizer hyperparameters | $\beta_1 = 0.9, \beta_2 = 0.999, eps = 1e - 8$ |
| Peak learning rate | 2e-5 |
| Learning rate schedule | cosine decay |
| Training epochs | 1 |
| Training steps | 5, 000 |
| Warm-up steps | 150 |
| Global batch size | 128 |
| Gradient accumulation | 2 |
| Numerical precision | bfloat16 |

While using Vlaser as the base model for downstream VLA Policy fine-tuning, we optimize all parameters within both the VLM and the Action Expert. Additionally, we conduct comparative experiments using several different versions of base models, including InternVL3-2B, etc. Detailed information and related parameter settings can be found in Table 7.

## A.2 DATA GENERATION DETAILS

**Embodied Grounding Data** To further enhance embodied grounding capabilities, we generate an additional 300k high-quality data samples from the SA-1B dataset (Kirillov et al., 2023). The

Table 7: **Hyper-parameters used in the VLA fine-tuning.**

| Configurations | Values |
|---|---|
| LLM sequence length | 384 |
| Image Size | 448 |
| Freeze VLM | False |
| Global batch size | 1024 |
| Training epochs | 10 |
| VLM Peak Learning Rate | 5e-5 |
| Action Expert Peak Learning Rate | 5e-5 |
| Optimizer | AdamW |
| Optimizer hyperparameters | $\beta_1 = 0.9, \beta_2 = 0.999, eps = 1e - 8$ |
| Observation history length | 1 |
| Action Chunk length | 4 |
| Execute Action length | 2 |

data generation process consists of two main stages. First, we convert segmentation masks into bounding boxes and point annotations: bounding boxes are derived by computing the minimal axis-aligned rectangle enclosing each mask, while point annotations are obtained by randomly sampling a coordinate within the mask region. To ensure annotation quality, we apply an IoU threshold of 0.9 to select high-precision masks; masks with lower IoU values are either excluded or assigned reduced sampling weight. From the over 1 billion available masks, we initially sample 1 million candidate instances. In the second stage, we employ a two-step captioning and refinement pipeline. Coarse captions are first generated using BLIP-2 (Li et al., 2023), followed by a filtering and refinement process using Qwen2.5-VL-7B (Bai et al., 2025) to eliminate low-quality items and produce more accurate and detailed descriptions. This rigorous pipeline ultimately yields 300k high-quality data samples tailored for embodied grounding tasks, significantly expanding the diversity and precision of our training corpus.

**Spatial Reasoning Data** To enhance spatial intelligence capabilities, we manually construct a dataset of 100k 3D spatial perception samples derived from ScanNet (Dai et al., 2017), ScanNet++ (Yesh-wanth et al., 2023), and ARKitScenes (Baruch et al., 2021). Following methodologies established in prior data engines (Deng et al., 2025; Fan et al., 2025), we utilize both the 3D point cloud and video sequences of each scene to construct a spatio-temporal scene graph. This graph encapsulates structural and semantic information such as overall scene dimensions, room center coordinates, object category counts, and precise 3D bounding boxes for every object instance. Based on this representation, we generate spatial reasoning questions that probe layout properties and inter-object relationships. These include queries about the object counts, absolute and relative distances, object and room sizes, relative directions between objects, and other spatial attributes, using the same question template in VSI-Bench (Yang et al., 2025b).

**Planning Data** To improve the model's ability to comprehend complex instructions and execute tasks with environmental feedback, we curate additional planning data within the Habitat simulator (Szot et al., 2021). Specifically, we initialize each planning task following the annotations of LLaRP (Szot et al., 2024), which specify both the task goals and the set of permissible actions. An LLM agent based on gpt-4o (OpenAI, 2025) is then deployed to roll out the task. During each rollout, we record the task instruction, the sequence of actions taken, and the environment's feedback, including observations and success signal for each action. Both the executed action trajectories and the corresponding feedback are retained. Only trajectories that successfully accomplish the task are included in the final training set, providing rich paired data of instructions, execution processes, and environment responses for enhancing the model's planning capabilities in a complex environment.

**In-Domain Data for downstream VLAs** We generate 2 million in-domain multimodal data samples to facilitate direct transfer learning for downstream Vision-Language Agents (VLAs) during fine-tuning. These data are collected from two distinct simulation platforms: the WidowX Robot (Walke et al., 2023a) and the Google Robot (Brohan et al., 2023b;a) within the SimplerEnv (Li et al., 2024d), as well as the Aloha-AgileX Robot from RoboTwin2.0 (Chen et al., 2025c). The dataset mainly encompasses three systematically designed question-answer types: 1) General QA, which queries the robot's current state and requests next few action plans; 2) Grounding QA, which requires the robot to localize points and bounding boxes as the actionable affordances; 3) Spatial Reasoning

QA, which probes relative spatial relationships and geometric properties of objects in the scene. Detailed prompt templates and representative examples for each data category are provided in Figure 3 (General QA), Figure 4 (Grounding QA), and Figure 5 (Spatial Reasoning QA), respectively. We use Qwen2.5VL-7B (Bai et al., 2025) as the base model to generate the above data items.

To further enhance the quality and diversity of the generated in-domain QA data, we implement a post-processing data filtering pipeline. In line with established practices in dataset construction (Wang et al., 2023; Cui et al., 2023; Li et al., 2024b), we employ an LLM-as-a-judge approach (Li et al., 2025) to score each generated data sample on a scale of 1 to 3, using Qwen2.5VL-32B (Bai et al., 2025) as the judge model. The detailed instruction prompts for General QA, Spatial QA, and Grounding QA are provided below. We filter out all samples assigned a score of 1, which account for approximately 10% of the initially generated data. This outcome reflects the overall high quality and diversity of our in-domain dataset. To further validate the reliability of the LLM-based evaluation, we randomly selected a subset of the scored data (covering all score levels from 1 to 3) for human reassessment by three human experts. The results show that the LLM judge's scores align with human ratings in nearly 80% of cases, confirming the consistency and reliability of our automated evaluation and data filtering pipeline.

---

### Prompt of Data filtering for General QA data

**You are an expert in evaluating question-answer pairs for robot arm camera images and task instructions. Please evaluate the quality of the generated QA pair based on Relevance, Informativeness, and Clarity.**

You will receive the following inputs: — The robot arm camera image, the task instruction, the generated question, and the generated answer. Carefully evaluate whether the question and answer are relevant to the image and task, informative, and clearly expressed.

**Scoring Criteria (1–3):**

- **1 – Poor** The QA pair is irrelevant, uninformative, or unclear. - The question or answer doesn't relate to the image or task instruction. - The answer is vague, generic, or provides no useful information. - The question is poorly formulated or doesn't make sense.

  **Examples:** - Question: "What is this?" Answer: "It's an image." (too generic) - Question: "How to cook?" Answer: "Use a pan." (irrelevant to robot task) - Question: "What color?" Answer: "Color." (unclear and uninformative)

- **2 – Fair** The QA pair is somewhat relevant but lacks depth or clarity. - The question relates to the image/task but is too simple or generic. - The answer provides basic information but lacks detail or specificity. - Some aspects are unclear or could be better explained.

  **Examples:** - Question: "What objects are in the image?" Answer: "There are some objects." (too vague) - Question: "What is the robot doing?" Answer: "The robot is moving." (lacks detail) - Question: "How to complete the task?" Answer: "Follow the instructions." (not specific enough)

- **3 – Good** The QA pair is highly relevant, informative, and clearly expressed. - The question is specific, well-formulated, and directly relates to the image and task. - The answer provides detailed, accurate, and useful information. - Both question and answer are clear and help understand the robot's environment and task.

  **Examples:** - Question: "What objects are visible in the robot arm's workspace and which one should be manipulated based on the task instruction?" Answer: "I can see a red cup, a blue bowl, and a green plate on the table. According to the task instruction to 'pick up the cup', the robot should focus on the red cup located in the center of the workspace." - Question: "What obstacles might prevent the robot from completing the task?" Answer: "The workspace appears clear, but the target object is positioned near the edge of the table, which may require careful positioning to avoid knocking it over during manipulation."

**Output Requirement:** You must return only a single integer score from 1 to 3. Do not include any explanation, labels, or extra content.
**Question:** {question}
**Answer:** {answer}

---

### Prompt of Data filtering for Spatial QA data

**You are an expert in evaluating spatial intelligence question-answer pairs for robot arm camera images and task instructions. Please evaluate the quality based on Spatial Reasoning Accuracy, Detail Level, and Relevance.**

You will receive the following inputs: — The robot arm camera image, the task instruction, the generated spatial question, and the generated spatial answer.
Carefully evaluate whether the question targets spatial reasoning, and whether the answer provides accurate and detailed spatial information.

**Scoring Criteria (1–3):**

- **1 – Poor** The spatial QA pair lacks spatial reasoning focus or provides incorrect/vague spatial information. - The question doesn't target spatial aspects (counting, relationships, distances, orientation, etc.). - The answer provides incorrect spatial information or is too vague. - The spatial reasoning is flawed or irrelevant to the task.

**Examples:** - Question: "What is in the image?" Answer: "Objects." (not spatial) - Question: "How many objects?" Answer: "Some." (vague, no specific count) - Question: "Where is object A?" Answer: "It's there." (no spatial detail) - Question: "What is the distance?" Answer: "Close." (not quantitative)

- **2 – Fair** The spatial QA pair addresses spatial reasoning but lacks precision or detail. - The question targets spatial aspects but could be more specific. - The answer provides basic spatial information but lacks quantitative details or precision. - Some spatial relationships are mentioned but not fully explained.

  **Examples:** - Question: "How many cups are there?" Answer: "There are a few cups." (not specific count) - Question: "Where is the red object relative to the blue one?" Answer: "The red one is near the blue one." (lacks specific direction/distance) - Question: "What is the spatial arrangement?" Answer: "Objects are arranged on the table." (too general)

- **3 – Good** The spatial QA pair demonstrates strong spatial reasoning with precise, detailed information. - The question clearly targets specific spatial aspects (counting, relationships, distances, orientation, geometry, etc.). - The answer provides accurate, quantitative, and detailed spatial information. - The spatial reasoning is relevant to the robot task and helps understand the workspace layout.

  **Examples:** - Question: "How many objects are visible in the scene and what are their types?" Answer: "I can count 5 objects: 2 red cups positioned on the left side of the table, 1 blue bowl in the center, and 2 green plates arranged on the right side." - Question: "What is the relative position of the target object compared to the robot arm's current position?" Answer: "The target object (red cup) is located approximately 30cm to the right and 15cm forward from the robot arm's current end-effector position, requiring a diagonal reach motion." - Question: "Which objects are within the robot's reachable workspace?" Answer: "Based on the robot arm's reach, the blue bowl (center, 25cm away) and the left red cup (20cm away) are within reach. The rightmost green plate (45cm away) is outside the immediate reachable zone."

**Output Requirement:** You must return only a single integer score from 1 to 3. Do not include any explanation, labels, or extra content.
**Question:** {question}
**Answer:** {answer}

---

## Prompt of Data filtering for Grounding QA data

**You are an expert in evaluating visual grounding question-answer pairs for robot arm camera images and task instructions. Please evaluate the quality based on Grounding Accuracy, Coordinate Validity, and Localization Precision.**

You will receive the following inputs: — The robot arm camera image, the task instruction, the generated grounding question, and the generated grounding answer (which should contain coordinates or bounding boxes). Carefully evaluate whether the question targets object localization, and whether the answer provides valid and accurate coordinate information.

**Scoring Criteria (1–3):**

- **1 – Poor** The grounding QA pair lacks localization focus or provides invalid/incorrect coordinate information. - The question doesn't target object localization, pointing, or detection. - The answer lacks coordinate information or contains invalid coordinates (out of bounds, wrong format). - The coordinates don't match the described object location.

  **Examples:** - Question: "What is in the image?" Answer: "A cup." (no coordinates) - Question: "Where is the object?" Answer: "It's on the table." (no coordinates) - Question: "Point to the cup." Answer: "The cup is at (1500, 2000)." (coordinates out of bounds for normalized 0-1000 range) - Question: "Find the object." Answer: "Box: [100, 200, 50, 300]." (invalid box format, x2 < x1)

- **2 – Fair** The grounding QA pair addresses localization but coordinates are imprecise or partially valid. - The question targets localization but could be more specific. - The answer contains coordinates but they may be approximate, slightly off, or lack precision. - Some coordinate information is present but formatting could be improved.

  **Examples:** - Question: "Where is the object?" Answer: "The object is at position (500, 300)." (single point, but should specify if multiple objects exist) - Question: "Point to all cups." Answer: "Cups are at (200, 150) and (600, 400)." (coordinates present but may not be precise) - Question: "Mark the boundaries." Answer: "Box: [100, 100, 400, 300]." (valid box but may not accurately bound the object)

- **3 – Good** The grounding QA pair demonstrates precise localization with valid and accurate coordinate information. - The question clearly targets object localization, pointing, detection, or precise positioning. - The answer provides valid, accurate coordinates (points or bounding boxes) in the correct format. - Coordinates are normalized to 0-1000 range and accurately represent object locations. - Multiple objects are properly localized if applicable.

  **Examples:** - Question: "Where is the red cup located in the image? Provide coordinates." Answer: "The red cup is located at point (450, 320). <point>[[450, 320]]</point>" - Question: "Point to all instances of cups visible in the scene." Answer: "I can locate 2 cups: the red cup at (450, 320) and the blue cup at (680, 250). <point>[[450, 320], [680, 250]]</point>" - Question: "Can you locate and mark the boundaries of the target object?" Answer: "The target object (red cup) is bounded by box [380, 280, 520, 360]. <box>[[380, 280, 520, 360]]</box>" - Question: "Find and mark all instances of plates in the image." Answer: "I found 2 plates: plate 1 at box [100, 200, 250, 350], plate 2 at box [600, 180, 750, 330]. <box>[[100, 200, 250, 350], [600, 180, 750, 330]]</box>"

**Output Requirement:** You must return only a single integer score from 1 to 3. Do not include any explanation, labels, or extra content.
**Question:** {question}
**Answer:** {answer}

## A.3 SIMULATION EVALUATION DETAILS

We fine-tune and evaluate the VLA models using various base models within the SimplerEnv simulation environment. To ensure fair evaluation, we use checkpoints with the same number of iterations for the WidowX Robot Task and the Google Robot Task, respectively. Specifically, for the WidowX Robot Tasks, we use a checkpoint after 45,390 iterations, while for the Google Robot Tasks, we use a checkpoint after 36,970 iterations. During evaluation, we utilize a single image and select an action chunk of size 2 for execution. In the flow matching configuration, we use 10 inference steps during the inference phase and apply Euler method as numerical integration method. We evaluate a sufficient number of samples to ensure the reliability of the tests. The exact number of test samples for each task is shown in the Table 8.

Table 8: **The number of samples evaluated on SimplerEnv.**

| | | Task Name | Evaluation Samples |
|---|---|---|---|
| Widowx Robot Tasks | | Carrot on plate | 240 |
| | | Put eggplant in basket | 240 |
| | | spoon on towel | 240 |
| | | stack cube | 240 |
| Google Robot Tasks | Visual Matching | Pick coke can | 300 |
| | | Move Near | 240 |
| | | Open/Close Drawer | 216 |
| | Variance Aggregation | Pick coke can | 825 |
| | | Move Near | 600 |
| | | Open/Close Drawer | 378 |

## A.4 QUALITATIVE DEMONSTRATION

**Qualitative Samples for Embodied Reasoning**. To gain in-depth insights into Vlaser, we provide extensive visualizations on embodied reasoning tasks including Embodied QA 9, grounding 10, spatial intelligence 11 and planning tasks in Simulation 12. These examples not only validate the generalizability of Vlaser, but also confirm the superior capabilities of Vlaser across all tasks.

**Qualitative Samples in SimplerEnv**. Figure 6 and Figure 7 present qualitative examples from the evaluations conducted in the SimplerEnv. Specifically, Figure 6 showcases the performance of multiple models on the WidowX Robot Task, while Figure 7 demonstrates the results on the Google Robot Task. From these results, it is evident that, under the same test conditions, baseline models often fail the entire task due to slight positional errors. In contrast, our model shows significant improvements in this regard. Whether incorporating in-domain question-answer pair data, spatial intelligence data, or grounding data, the performance of the model shows clear enhancement. These results validate the effectiveness of our approach.

**General QA**

**Instruction**: You are an AI assistant analyzing robot arm camera images and task instructions.
Given the robot arm camera image and the task instruction: "{text}"
Please generate a natural question-answer pair about this image and task. The question should be open-ended and could ask about:
- Objects visible in the image - The robot arm's current state or position
- How to accomplish the given task - What obstacles or challenges might exist
- Safety considerations for the task - Or any other relevant aspect of the image and task
Please respond in the following format:
Question: [Your question here] Answer: [Your detailed answer here]
Make sure the question is natural and the answer is informative and helpful for understanding the robot arm's environment and task.

**Question**: How can the robot arm effectively fold the green cloth from left to right as instructed?
**Answer**: To fold the green cloth from left to right, the robot arm should first position itself so that the left edge of the cloth is aligned with the starting point of the fold. The arm should then move in a controlled manner, applying gentle pressure to fold the cloth over to the right edge. It's important to ensure that the cloth is not torn or damaged during the folding process. The robot arm should also be equipped with sensors to detect the cloth's texture and adjust its movements accordingly to maintain a smooth and accurate fold. Additionally, the robot arm should be programmed to stop folding once the right edge is reached, ensuring that the cloth is neatly folded without any wrinkles or creases.

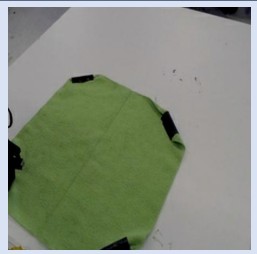

*WidowX Robot*

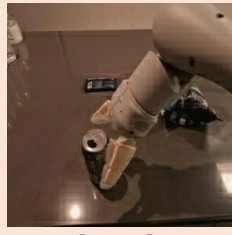

*Google Robot*

**Question**: What safety measures should be taken when using a robot arm to pick up cans like the can shown in the image?
**Answer**: When using a robot arm to pick up cans, several safety measures should be considered to ensure the task is performed safely and efficiently. First, it's crucial to ensure that the robot arm is properly calibrated and that the gripper is securely attached to the arm. This prevents accidental drops or damage to the can. Additionally, the robot arm should be programmed with gentle and precise movements to avoid any impact that could damage the can or the arm itself. It's also important to ensure that the robot arm is operating within its designated safety boundaries to prevent any unintended movements that could cause harm. Furthermore, the environment should be free of obstacles that could interfere with the robot arm's movement, and the area should be well-lit to ensure the robot arm can accurately identify and grasp the can. Lastly, it's advisable to have a human supervisor or safety system in place to monitor the robot arm's actions and intervene if necessary.

**Question**: What is the current state of the robot arm and how can it accomplish the task of lifting the pot?
**Answer**: The robot arm is currently positioned above the pot, with its grippers engaged around the pot's handle. To accomplish the task of lifting the pot, the robot arm should ensure a secure grip on the handle and then apply upward force to lift the pot off the surface. The robot arm should also be mindful of any potential obstacles or uneven surfaces that could interfere with the lifting process, and maintain a steady and controlled motion to avoid spilling the contents of the pot. Safety considerations include ensuring that the robot arm is properly calibrated and that the pot is within the weight capacity of the grippers. Additionally, the robot arm should be aware of any nearby objects or people that could be at risk of being hit or injured during the lifting process. Overall, the robot arm should prioritize precision, control, and safety to successfully complete the task of lifting the pot.

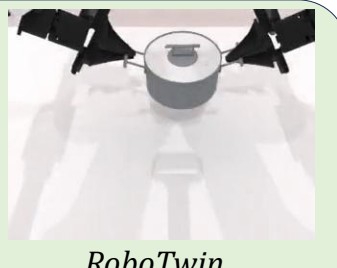

*RoboTwin*

Figure 3: An illustration of Vlaser-6M data engine for in-domain general QA sample in SimplerEnv and RoboTwin.

## Grounding QA

**Instruction** : You are an AI assistant specializing in visual grounding analysis of robot arm camera images and task instructions.
Given the robot arm camera image and the task instruction: "{text}"
Please generate a natural question-answer pair specifically focused on visual grounding capabilities. The question should target object localization and could ask about:
- Object pointing: "Where is the [specific object] located in the image?"
- Multiple object pointing: "Point to all the [objects] visible in the scene."
- Object detection: "Can you locate and mark the boundaries of the [object]?"
- Multiple object detection: "Find and mark all instances of [objects] in the image."
- Spatial localization: "Where exactly can we find the [object] that the robot needs to interact with?"
- Region identification: "Which area of the image contains the [target object]?"
- Precise positioning: "What are the exact coordinates of the [object] in the image?"

You should either return a set of 2D points or a set of 2D bounding box(es) as the answer for the specific visual grounding question.
Please respond in the following format:
Question: [Your grounding question here]
Answer: [Your detailed localization answer with specific coordinate information]

The point format could be like the following:
coordinate_patterns = [
    r'\((\d+(?:\.\d+)?),\s*(\d+(?:\.\d+)?)\)',  # (x, y)
    ]
The box format could be like the following:
box_patterns = [
    r'\[(\d+(?:\.\d+)?),\s*(\d+(?:\.\d+)?),\s*(\d+(?:\.\d+)?),\s*(\d+(?:\.\d+)?)\]',  # [x1, y1, x2, y2]
    ]

Make sure the question focuses on object localization, positioning, or boundary detection, and the answer provides precise location information about objects in the robot's visual field.

### WidowX Robot

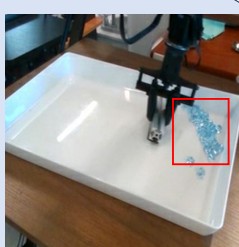

**Question**: Where is the pile of small blue objects located in the image? Your answer should be formatted as \"<box>[[x1, y1, x2, y2], [x3, y3, x4, y4], ...]</box>\". The bounding box coordinates are normalized to integers between 0 and 1000. Return the answer in the bounding box format directly.
**Answer**: <box>[[761, 364, 912, 608]]</box>

### Google Robot

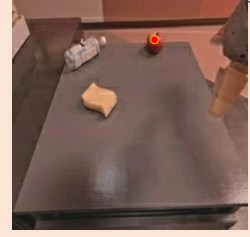

**Question**: Where is the apple located in the image? Your answer should be formatted as \"<point>[[x1, y1], [x2, y2], ...]</point>\". The point coordinates are normalized to integers between 0 and 1000. Return the answer in the point format directly.
**Answer**: <point>[[612, 223]]</point>

### RoboTwin

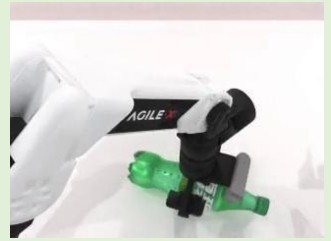

**Question**: In the visual input, can you identify the smooth green plastic bottle and determine its precise location on the table? Your answer should be formatted as \"<point>[[x1, y1], ...]</point>\". The point coordinates are normalized to integers between 0 and 1000.
**Answer**: <point>[[565, 775], [565, 775]]</point>"}]}

Figure 4: An illustration of Vlaser-6M data engine for in-domain embodied grounding QA sample in SimplerEnv and RoboTwin.

## Spatial Reasoning QA

**Instruction**: You are an AI assistant specializing in spatial intelligence analysis of robot arm camera images and task instructions.

Given the robot arm camera image and the task instruction: "{text}"

Please generate a natural question-answer pair specifically focused on spatial intelligence aspects. The question should target spatial reasoning and could ask about:
- Object counting: "How many [objects] are visible in the scene?"
- Spatial relationships: "What is the relative position of [object A] compared to [object B]?"
- Distance estimation: "How far apart are [object A] and [object B]?"
- Spatial orientation: "In which direction is [object] oriented relative to the robot arm?"
- Depth perception: "Which object appears closer/farther from the camera?"
- Geometric properties: "What is the approximate shape/size of [object] in the scene?"
- Spatial arrangement: "How are the objects arranged in the workspace?"
- Accessibility analysis: "Which objects are within the robot's reach based on their positions?"
- Collision avoidance: "What spatial constraints must the robot consider when moving toward [target object]?"
- 3D spatial understanding: "What is the vertical/horizontal relationship between objects?"

Please respond in the following format:
Question: [Your spatial intelligence question here]
Answer: [Your detailed spatial analysis answer here]

Make sure the question focuses on spatial reasoning, object relationships, positioning, or geometric understanding, and the answer provides precise spatial information about the robot's environment.

### WidowX Robot

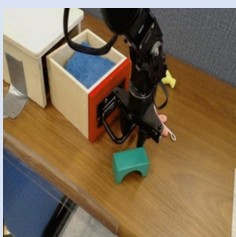

**Question**: What is the relative position of the drawer compared to the robot arm?
**Answer**: The drawer is positioned directly beneath the robot arm, with the arm's gripper currently interacting with the surface of the drawer. The drawer is oriented horizontally, and its front edge is aligned with the front edge of the robot arm's gripper. The drawer appears to be slightly tilted forward, suggesting that the robot arm is in the process of opening it.

### Google Robot

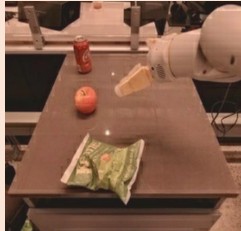

**Question**: What is the relative position of the coke can compared to the apple on the table?
**Answer**: The coke can is positioned to the left of the apple on the table. The can is closer to the top left corner of the table, while the apple is slightly below and to the right of the can.

### RoboTwin

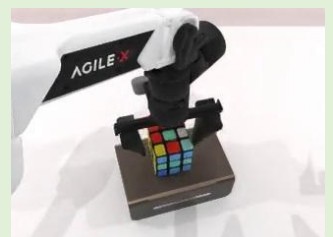

**Question**: Is the Rubik's cube currently held by the robot arm positioned above the display stand with a rectangular hollow base, and is there sufficient clearance for the cube to be placed onto the stand without obstruction?
**Answer**: Yes, the Rubik's cube is held above the display stand, and there is sufficient clearance for the cube to be placed onto the stand without obstruction. The cube is aligned with the rectangular hollow base of the stand, and the robot arm has enough space to lower the cube into the base without any visible obstacles in the immediate vicinity. The arm's position and the cube's orientation suggest that the placement can be executed smoothly and precisely.

Figure 5: An illustration of Vlaser-6M data engine for in-domain spatial reasoning QA sample in SimplerEnv and RoboTwin.

Table 9: Embodied QA examples.

**Embodied QA: Example #1 from ERQA.**

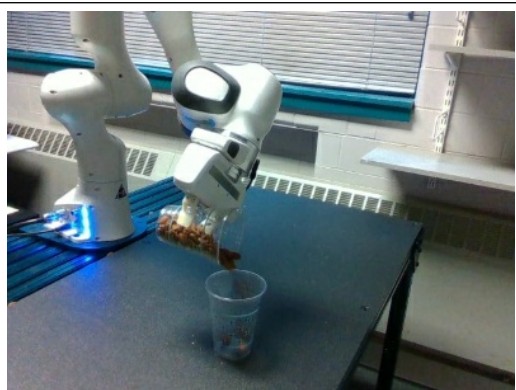 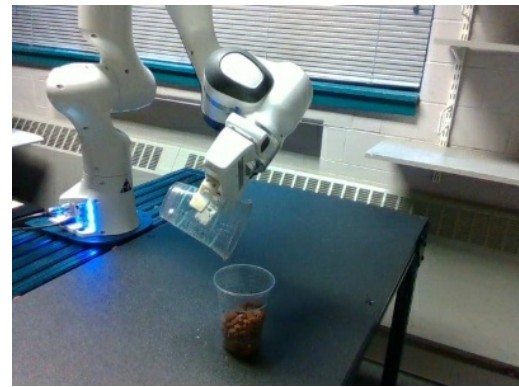

| **Question**: | What happened between these two frames? Choices: A. Robot arm lifted the cup. B. Robot arm poured all the nuts into a cup. C. Robot arm poured some of the nuts into a cup. D. Nothing happened. Please answer directly with only the letter of the correct option and nothing else. |
|---|---|
| **Vlaser-8B**: | C. |

**Embodied QA: Example #2 from ERQA.**

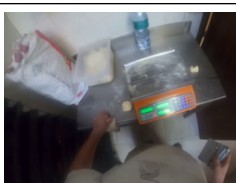 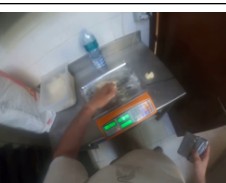 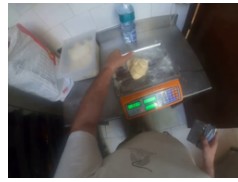 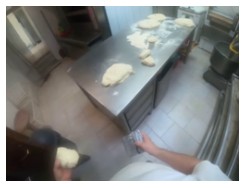

| **Question**: | Select the best answer to the following multiple-choice question based on the video. Respond with only the letter (A, B, C, or D) of the correct option. Considering the progress shown in the video and my current observation in the last frame, what action should I take next in order to weigh dough? A. put dough on table B. pick dough C. roll dough on table D. cut dough with dough cutter |
|---|---|
| **Vlaser-8B**: | A. |

Table 10: Embodied Grounding Examples.

| **Embodied Grounding: Example #1 from PointArena.** | |
| --- | --- |

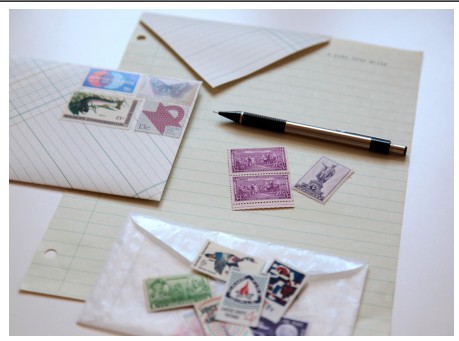 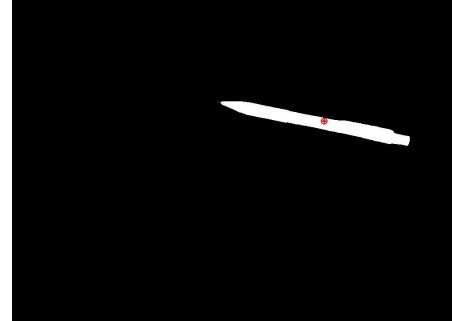

| **Question**: | Point to the tool people use for writing. |
| --- | --- |
| **Vlaser-8B**: | <point>[[701, 374]]</point>. |

| **Embodied Grounding: Example #2 from Where2Place.** | |
| --- | --- |

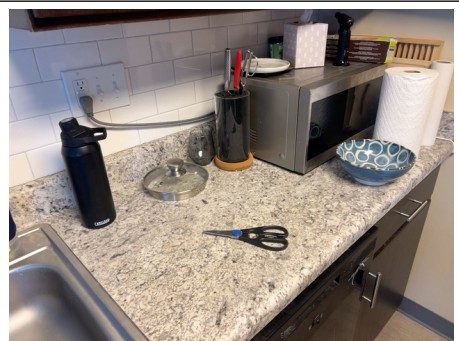 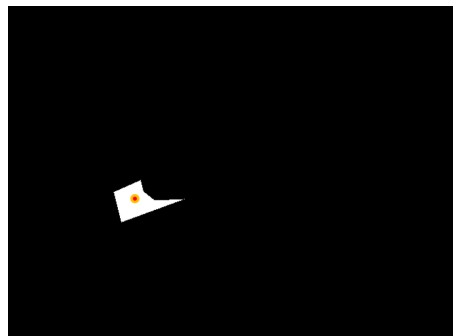

| **Question**: | Please point out the free space between the black water bottle and the pot lid. |
| --- | --- |
| **Vlaser-8B**: | <point>[[293, 560]]</point>. |

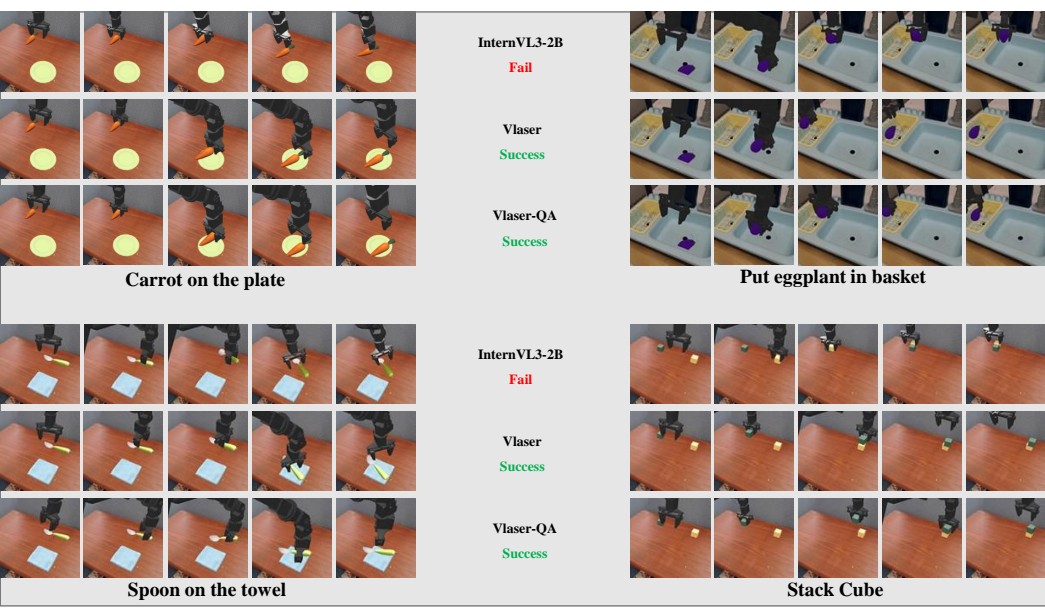

Figure 6: Qualitative samples in SimplerEnv on WidowX Robot Tasks

Table 11: Spatial Intelligence Examples.

**Spatial Reasoning: Example #1 from VSI.**

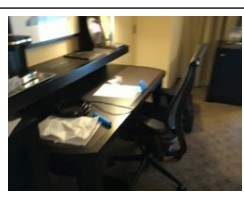 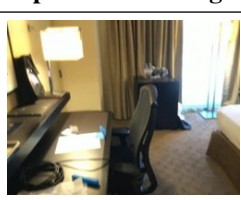 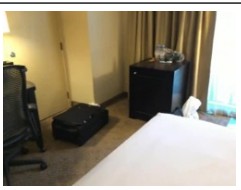 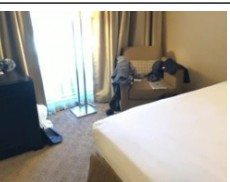

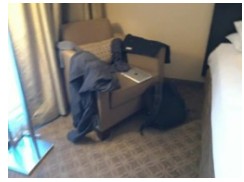 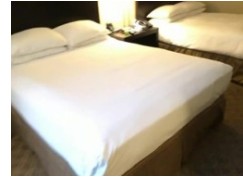 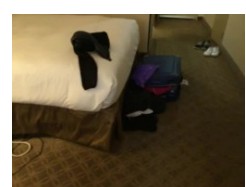 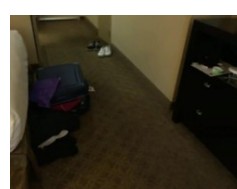

| | |
|---|---|
| **Question**: | You are a robot beginning at the standing by the window and facing the window. You want to navigate to the white shoes. You will perform the following actions (Note: for each [please fill in], choose either 'turn back,' 'turn left,' or 'turn right.'): 1. [please fill in] 2. Go forward passing the bed 3. [please fill in] 4. Go forward until the white shoes. You have reached the final destination. A. Turn Right, Turn Left B. Turn Left, Turn Left C. Turn Left, Turn Right D. Turn Right, Turn Right. Answer with the option's letter from the given choices directly. |
| **Vlaser-8B**: | B. |

Table 12: Simulation planning Examples.

| Simulation Planning: Example #1 from EmbodiedBench. |
|---|

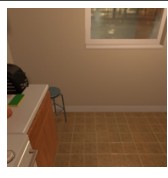 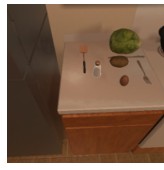 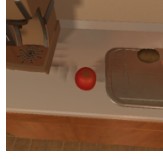 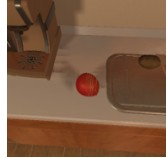 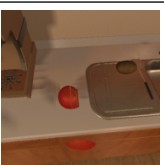

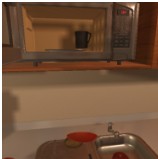 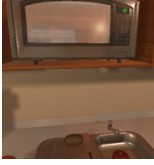 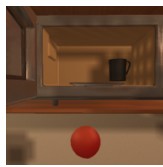 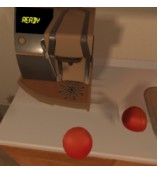 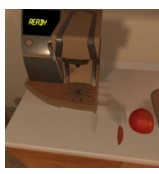

**Question**: You are a robot operating in a home. Given a task, you must accomplish the task using a defined set of actions to achieve the desired outcome. Now the human instruction is: Put a microwaved tomato slice on the counter. You need to output the reasoning process and an action to execute.

**Vlaser-8B**: Reasoning-step-1: To put a microwaved tomato slice on the counter, I need to start by slicing a tomato. To do this, I require a knife. In the image, there is a knife visible on the counter near the toaster. Finding the knife is the logical next step because it is essential for cutting the tomato into slices. Once I have the knife, I can proceed with slicing the tomato. Therefore, my next action is to find the knife. Action: find a knife.

Reasoning-step-2: I have identified the knife on the counter near the sink. Picking up the knife is the next logical step because it is necessary for slicing the tomato. Once I have the knife, I can proceed with cutting the tomato into slices. This aligns with the goal of preparing a tomato slice for microwaving and then placing it on the counter. Action: pick up the knife.

Reasoning-step-3: I have successfully located the knife in the drawer. Now, I need to find a tomato to slice. The image shows a kitchen counter with various items, but no tomato is immediately visible. Tomatoes are commonly stored in the refrigerator or on the counter. Since the refrigerator is partially visible on the left side of the image, it makes sense to check there first. If the tomato isn̈'t in the fridge, I can then look around the counter more thoroughly. Therefore, my next step is to find a tomato. Action: find a tomato.

...

Reasoning-step-9: The current visual state shows the microwave with its door closed, and the tomato slice is on the counter. The task is to place a microwaved tomato slice on the counter, which has already been accomplished. Given that the tomato slice is already on the counter, the next logical step is to confirm its placement by finding a suitable counter top. This ensures that the tomato slice is securely and appropriately positioned for the task. Therefore, the next step is to find a counter top. Action: find a counter top.

Reasoning-step-10: The current visual state shows a microwave with its door closed, and two tomatoes are visible on the counter. One tomato is already sliced, indicating that the previous steps involving slicing and microwaving have been completed. Since the tomato is currently being held, the next logical step is to put it down on the counter. This action will free up the hand to handle the tomato slice and place it on the counter as required by the task. Therefore, the appropriate next action is to put down the tomato. Action: put down the tomato.

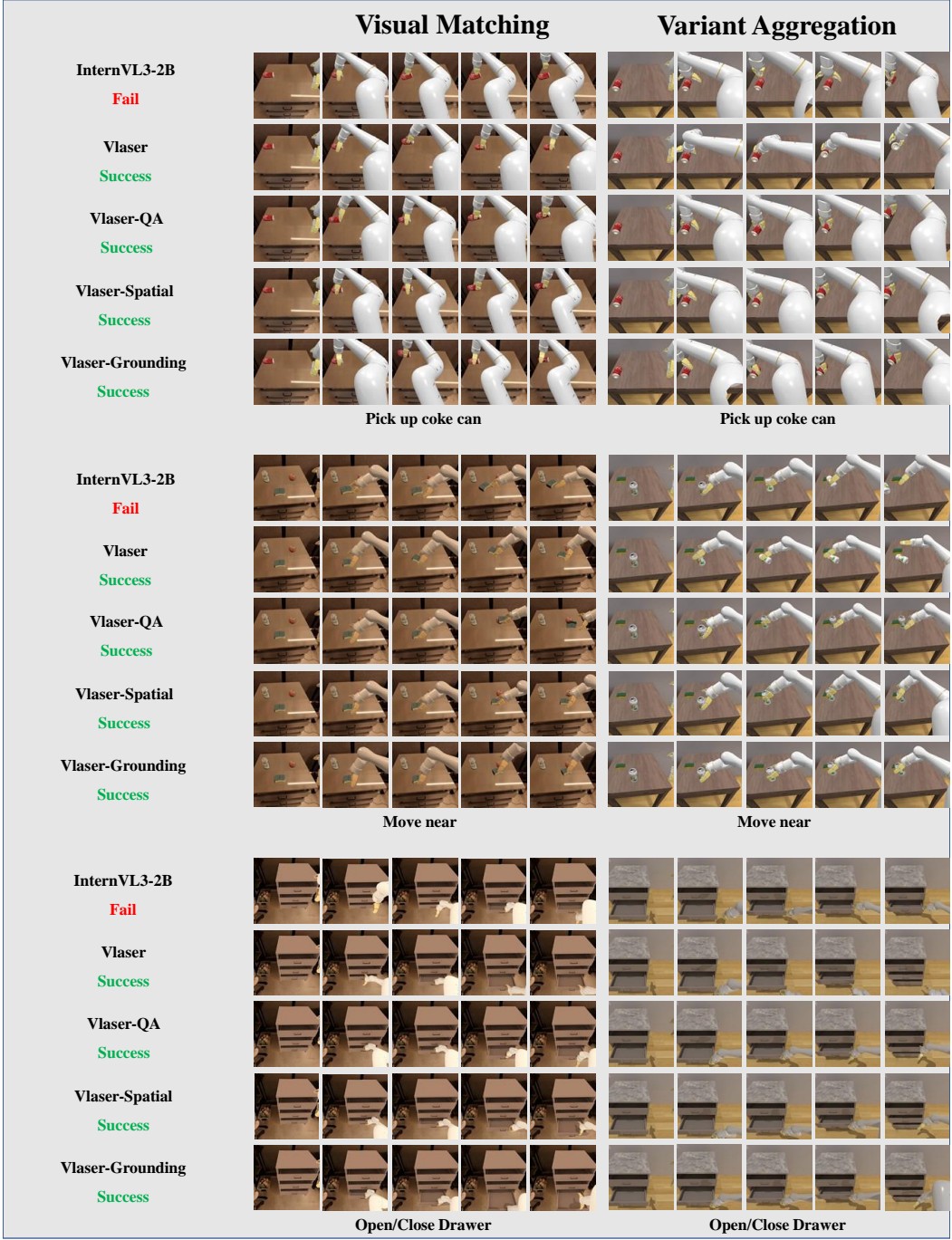

Figure 7: Qualitative samples in SimplerEnv on Google Robot Tasks

