# OpenReview forum: "Vlaser: Vision-Language-Action Model with Synergistic Embodied Reasoning"
_ICLR.cc/2026/Conference — ICLR 2026 Poster_

### Official Review · Reviewer_HWMv · 2025-10-27

**Soundness:** 2
**Presentation:** 3
**Contribution:** 2
**Rating:** 4
**Confidence:** 4

**Summary:**

This paper proposes VLASER, a VLA model with collaborative embodied inference ability, which aims to solve the domain gap between upstream VLM inference and downstream VLA strategy learning. VLASER is based on InternVL3, including a two-stage training process of "VLM pre-training + VLA fine-tuning", and relies on the Vlaser-6M dataset to achieve SOTA performance on 12 embodied inference benchmarks (Vlaser-8B average score of 51.3), while in WidowX Achieve the best results on the robot benchmark (Vlaser-QA average success rate of 64.6%) and perform competitively on the Google Robot benchmark; The study also reveals key findings: out-of-domain embodied inference data has limited VLA performance improvement, while in-domain data based on robot interaction data can accelerate VLA convergence and improve task success rate.

**Strengths:**

See Summary

**Weaknesses:**

1. In Section 3.2, the authors said there is no positive correlation between common embodied reasoning benchmarks and the closed-loop control performance of robot-specific embodied subordinates, but there is a lack of corresponding quantitative analysis or visualization support.
2. Figure 2 aims to show the two-stage training process of Vlaser, but the data flow and processing logic within each stage are not clear enough, and the relationship between the steps is unclear.
3. The description of the training process in Section 2.3 is fuzzy. Although the authors describe the two-stage training strategy and the form of the core loss function, there is not enough detail about the mathematical mechanism of key processes such as flow matching action generation. The relevant parameters only give fixed values, but do not explain their physical meaning or selection basis.
4. The paper mentions that the VLA data in 2 million domains is generated by Qwen2.5VL-7B, but does not explain how to control the data quality after the generation, and suggests supplementary data quality evaluation methods.
5. The paper points out that "in-domain data improves VLA performance", but does not clarify the separate contributions of general QA, grounding QA, and spatial inference QA in in-domain data.
6. The title of the paper mentions synergistic embodied reasoning, but it is not clear where the synergy is reflected.

**Questions:**

See Weakness

---

> ### Author Response · Authors · 2025-11-24
> **To Reviewer HWMv(Part 1)**
>
> **Q1:** There is a lack of corresponding quantitative analysis to support that there is no positive correlation between common embodied reasoning benchmarks and the closed-loop control performance.
>
> **A1:** Thank you for this valuable suggestion. Firstly, we would like to clarify that in the original Table 3, the entry "Vlaser(2B)" (last fourth from the bottom) with scores of 68.7% in visual matching and 51.3% in visual aggregation **is actually the Vlaser-2B model fine-tuned solely on Vlaser-6M out-of-domain (OOD) data specific for the embodied reasoning benchmarks in Sec. 3.1, without any in-domain data in Vlaser-6M dataset**. Its performance is indeed on par with the baseline model InternVL3-2B, which aligns with the statement in lines 412-413 of the original manuscript that *there is no positive correlation between common embodied reasoning benchmarks and the performance of closed-loop control of the lower level for the specific embodiment of the robot*. The performance improvements of our VLAs actually stem from fine-tuning the Vlaser-2B model on in-domain generated data-whether general QA, spatial QA, or grounding QA (as shown in the last three rows of Table 3). We apologize for the lack of clarity and any misunderstanding this may have caused. In the revised manuscript, we have updated the relevant parts of Section 3.2 and replaced "Vlaser" with "Vlaser-OOD" to clearly denote the out-of-domain model. The quantitative comparisons of **Vlaser-OOD** and baseline **InternVL3-2B** in revised Table 2, Table 3 and Table 4 further backup our above conclusion. We also present the results for Google Robot (Table 3) below. We hope these revisions resolve the confusion and better convey the contribution of our approach.
>
> | Model                 | Visual Matching   | Visual Matching | Visual Matching | Visual Matching | Variant Aggregation | Variant Aggregation | Variant Aggregation | Variant Aggregation |
> |----------------------|------------------------|--------------------|-----------------|--------|------------------------|--------------------|-----------------|--------|
> |                      | **Pick Coke Can** | **Move Near** | **Drawer** | **Avg** | **Pick Coke Can** | **Move Near** | **Drawer** | **Avg** |
> | InternVL3-2B         | 94.3%                | 78.8%             | 19.0%          | 64.0% | 80.4%                 | 72.7%             | 11.1%          | 54.7% |
> | Vlaser-OOD(2B)   | 85.0%                 | 76.3%             | 44.9%          | 68.7% | 74.4%                 | 69.2%             | 10.3%          | 51.3% |
> | Vlaser-QA (2B)  | 90.0%                 | 84.2%             | 44.4%          | 72.9% | 78.2%                 | 78.2%             | 13.0%          | 56.4% |
> | Vlaser-Spatial (2B) | 83.0%                 | 77.9%             | 56.0%          | 72.3% | 77.7%                 | 73.2%             | 13.2%          | 54.7% |
> | Vlaser-Grounding (2B) | 83.3%                 | 83.3%             | 54.2%          | 73.6% | 81.2%                 | 76.8%             | 17.0%          | 58.3% |
> | Vlaser-All (2B) | 91.0%                 | 85.4%             | 52.1%          | 76.2% | 80.5%                 | 77.7%             | 18.8%          | 59.0% |

---

> ### Author Response · Authors · 2025-11-24
> **To Reviewer HWMv(Part 2)**
>
> **Q2:** In Figure 2, the data flow and processing logic within each stage are not clear enough, and the relationship between the steps is unclear.
>
> **A2:** Thank you for this helpful feedback. We have thoroughly revised Figure 2 in the updated manuscript to better illustrate the pipeline and data flow. As shown in the new figure, in Stage 1 (VLM Pretraining), the model is trained on diverse embodied reasoning data - including general QA, planning, grounding, and spatial reasoning, using standard next-token-prediction loss. In Stage 2 (VLA End-to-End Fine-tuning), we introduce a separate action expert module alongside the pretrained VLM, forming a complete VLA architecture. This stage is trained with flow-matching loss on robot-specific control data. The pretrained VLM from Stage 1 serves as weight initialization for Stage 2, ensuring effective knowledge transfer while maintaining modularity. We believe the revised figure and accompanying description now more clearly convey the structural relationship and data flow between the two stages.
>
> **Q3:** Section 2.3 lacks detail about the mathematical mechanism of key processes such as flow matching action generation and the physical meaning of relevant parameters.
>
> **A3:** Thank you for this constructive feedback. We have expanded Section 2.3 "Vision-Language-Action Fine-tuning" to provide a more detailed explanation of the flow matching mechanism and the associated mathematical formulation. Following established flow-matching VLA methods [1,2], we now include a clearer description of the action expert's network architecture. We also elaborate on the flow matching objective: during training, the network learns to output $\mathbf{v}_{\theta}(\mathbf{A}^\tau_t, \mathbf{o}_t)$ to match the denoising vector field defined as $\mathbf{u}(\mathbf{A}^\tau_t | \mathbf{A}_t) = \epsilon - \mathbf{A}_t$, where $\epsilon \sim \mathcal{N}(\mathbf{0}, \mathbf{I})$. During inference, actions are generated by integrating the learned vector field from $\tau=0$ to $\tau=1$, starting from Gaussian noise $\mathbf{A}^0_t \sim \mathcal{N}(\mathbf{0}, \mathbf{I})$ and iterating as $\mathbf{A}^{\tau + \delta}_t = \mathbf{A}^{\tau}_t + \delta \mathbf{v}_{\theta}(\mathbf{A}^{\tau}_t, \mathbf{o}_t)$. We believe the updated Section 2.3 now contains all the necessary details and explanations for flow-matching mechanism of the training and inference process.
>
> **Q4:** The paper does not explain how to control the data quality for the in-domain 2 million VLA data, the reviewer suggests supplementary data quality evaluation methods.
>
> **A4:** Thank you for this constructive feedback. In response, to further enhance the quality and diversity of the generated in-domain QA data, as recommended by the reviewer, we additionally implement a data filtering pipeline as the post-processing process. Specifically, we adopt Qwen2.5VL-32B as judgement to score each generated data sample on a scale of 1 to 3 from the worst to best, while we filter out all samples assigned a score of 1, which account for approximately 10% of the initially generated data. The detailed data-filtering process and instruction prompts have also been added to the revised Appendix A.2 "In-Domain Data for downstream VLAs", and we use the refined in-domain data after filtering to train the newly-added VLA models presented in updated Table 2, Table 3 and Table 4.
>
> **Q5:** The paper does not clarify the separate contributions of general QA, grounding QA, and spatial inference QA in in-domain data.
>
> **A5:** Thank you for this constructive feedback. We agree that there needs to be more ablation studies to clarify the separate contributions of each type of in-domain data. As recommended by the reviews, we have additionally include more separate results for Vlaser-2B model fine-tuned with different types of in-domain data - general QA, spatial, and grounding, referred to as **Vlaser-QA**, **Vlaser-Spatial**, and **Vlaser-Grounding**, respectively, along with the combined version **Vlaser-All**. These additions can be found in the revised Table 2, Table 3, and Table 4 of the updated manuscript (Note that Vlaser-QA,
> Vlaser-Spatial, and Vlaser-Grounding were already compared in the original Table 3 for Google Robot). We could conclude that incorporating any single type of in-domain data during pretraining leads to significant performance gains in VLA task execution, while combining all three data types yields further improvement. We also present the results for Google Robot (Table 3) above. We hope these expanded experiments will further help clarify the individual contributions of each in-domain data type and facilitate better reproducibility.

---

> ### Author Response · Authors · 2025-11-24
> **To Reviewer HWMv(Part 3)**
>
> **Q6:** It is not clear where the "synergy" in the title is reflected.
>
> **A6:** Thanks for the valuable question. As we mentioned in the Abstract, in this work we take an initial step toward bridging embodied reasoning with VLA policy learning. The word "synergy" comes from two aspects. *1) Synergy Between Pre-training and Fine-tuning.* Our experiments demonstrate that pretraining on in-domain embodied reasoning datas, whether general QA, spatial QA, or grounding QA, achieves substantial performance improvements (Vlaser-QA/Spatial/Grounding/All). So we could say the pretraining stage actually "synergizes" the subsequent VLA finetuning. *2) Synergy Between Data Types and Capabilities.* Our ablation studies (in the new Table 2-4 and Section 3.2) show that the combination of different in-domain data types (general QA, spatial, grounding) during pre-training leads to greater performance gains than any single type alone (as seen in the Vlaser-All results). This indicates that these reasoning capabilities are not isolated, they interact and "synergize" to produce a more robust and generalizable VLA policy.
>
> [1] Kevin Black, Noah Brown, Danny Driess, Adnan Esmail, Michael Equi, Chelsea Finn, Niccolo Fusai, Lachy Groom, Karol Hausman, Brian Ichter, et al. pi_{0}: A vision-language-action flow model for general robot control. arXiv preprint arXiv:2410.24164, 2024.
>
> [2] Physical Intelligence, Kevin Black, Noah Brown, James Darpinian, Karan Dhabalia, Danny Driess, Adnan Esmail, Michael Equi, Chelsea Finn, Niccolo Fusai, et al. pi_{0.5}: a vision-language-action model with open-world generalization. arXiv preprint arXiv:2504.16054, 2025.

---

> ### Author Response · Authors · 2025-11-27
> **Thanks for raising the score**
>
> We are very appreciated that you have increased your rating of this paper. We truly appreciate your professional and kind feedback. we are thankful for the acknowledgment of the novelty and contribution of our work, as well as the acknowledgment of the additional experiments and paper revisions we incorporated during the rebuttal period.

---

### Official Review · Reviewer_ZtrG · 2025-10-27

**Soundness:** 3
**Presentation:** 3
**Contribution:** 3
**Rating:** 6
**Confidence:** 3

**Summary:**

This paper introduces an embodied vision-language-action model built on InternVL together with the Vlaser-6M data engine. It uses a two-stage recipe, embodied VLM pretraining followed by VLA fine-tuning with a flow-matching action expert for low-level control. The paper provides an analysis of which pretraining data streams transfer best to VLA policy learning, giving practical guidance on building task-aware data mixtures. Vlaser attains state-of-the-art results across embodied reasoning benchmarks and shows competitive closed-loop manipulation performance.

**Strengths:**

* The paper is clearly written and easy to follow.
* This paper tackles an important problem: bridging VLMs and VLAs via embodied-data pretraining and in-domain fine-tuning.
* Results on embodied reasoning benchmarks and SimplerEnv are promising.

**Weaknesses:**

* For a more informative study, I suggest the authors to include ablations on the pretaining dataset, quantifying how each source contributes to embodied reasoning and robot control.

* In the planning data curation, the authors collected data from Habitat. Did you avoid overlap with the EB-Habitat evaluation set? Additionally, please elaborate more on why Habitat was chosen for planning data over alternatives such as ALFRED.

* For in-domain data curation, how are the questions and answers in the QA/Grounding/Spatial pairs generated? How to ensure the quality and diversity of the dataset?

* In the training recipe, the authors adopt action chunks for robot control. An ablation over different chunk sizes would help motivate this design.

* In Table 3, I do not see results for combining all in-domain data, yet line 419 claims gains from this combination. Please add the result or clarify.

* A further limitation is the lack of real-robot experiments. While challenging, this should at least be acknowledged in the limitations section.

**Questions:**

Please refer to the weakness part.

---

> ### Author Response · Authors · 2025-11-24
> **To Reviewer ZtrG(Part 1)**
>
> **Q1:** The reviewer suggest the authors to include ablations on the pretraining dataset to quantify each data source's contributions to embodied reasoning and robot control.
>
> **A1:** Thank you for your valuable suggestion. In response to embodied reasoning, we have added ablation studies analyzing the impact of different types of embodied reasoning data - namely embodied grounding, general QA, spatial reasoning, and planning data, as shown in the table below. The results reveal a clear positive correlation between data composition and model performance: incorporating the task-specific data (like planning data) significantly improves performance on the corresponding benchmarks (embodied planning benchmarks), without substantially affecting other types of embodied reasoning capabilities (like embodied grounding capability). This finding aligns with established conventions in multimodal learning research.
>
> |                                | QA       | Embodied Grounding | Embodied Grounding | Embodied Grounding | Spatial Intelligence | Spatial Intelligence | Planning & Simulation | Planning & Simulation | Planning & Simulation | Avg     |
> | ------------------------------ | -------- | ------------------ | ------------------ | ------------------ | -------------------- | -------------------- | --------------------- | --------------------- | --------------------- | ------- |
> | **Model**                      | **ERQA** | **Where2place**    | **Pointarena**     | **Paco-Lavis**     | **VSIBench**         | **RefSpatial**       | **Ego-Plan2**         | **VLABench**          | **EB-ALFRED**         | **avg** |
> | InternVL3-2B                   | 31.5     | 5.2                | 7.1                | 15.4               | 31.5                 | 1.8                  | 30.9                  | 19.4                  | 1.3                   | 16.0    |
> | + 1.8M Embodied Grounding Data | 31.5     | 79.2               | 60.3               | 80.1               | 27.9                 | 20.2                 | 15.4                  | 16.2                  | 0.2                   | 36.8    |
> | +  1.2M General QA Data        | 36.8     | 78.8               | 61.1               | 74.2               | 28.8                 | 23.5                 | 22.3                  | 20.1                  | 20.1                  | 40.6    |
> | + 600k Spatial Reasoning Data  | 36.2     | 75.1               | 55.5               | 74.4               | 59.8                 | 42.7                 | 21.8                  | 18.7                  | 21.3                  | 45.1    |
> | + 400k Planning Data           | 35.8     | 74.0               | 57.8               | 72.5               | 57.5                 | 43.0                 | 38.3                  | 23.1                  | 42.3                  | 49.4    |
>
> Regarding robot control, we mainly analyze the individual contributions of different data types within the 2 million in-domain data samples used for downstream VLAs - specifically, embodied QA (including planning), embodied spatial intelligence, and embodied grounding data. The corresponding Vlaser-2B models fine-tuned on each in-domain data type are denoted as **Vlaser-QA**, **Vlaser-Spatial**, and **Vlaser-Grounding**, respectively, along with a combined version **Vlaser-All**. These results are included in the revised Table 2, Table 3, Table 4, and Section 2.3 of the updated manuscript. (Note that
> Vlaser-QA, Vlaser-Spatial, and Vlaser-Grounding were already compared in the original Table 3 for Google Robot.) The results indicate that incorporating any type of in-domain data during pretraining leads to significant performance gains in VLA task execution, while combining all data types yields further improvement.

---

> ### Author Response · Authors · 2025-11-24
> **To Reviewer ZtrG(Part 2)**
>
> **Q2:** Please elaborate more on why Habitat was chosen for planning data over alternatives such as ALFRED, and did the author avoid overlap with the EB-Habitat evaluation set.
>
> **A2:** Firstly, there is no overlap with the EB-Habitat evaluation set. We only use the training split of LLaRP [1] to construct Habitat tasks for planning data. EB-Habitat evaluates solely on its test split, which contains complex and long-horizon tasks. We verified that there is no overlap between our collected data and the EB-Habitat evaluation set, and we'll release the corresponding collected data as part of our Vlaser-6M dataset.
>
> Secondly, Habitat provides large-scale and diverse tasks suitable for generating a wide range of planning trajectories. ALFRED is also valuable, as shown in Section 2.2, we adopt the reasoning-enhanced ALFRED trajectories from WAP [2], convert them into a planning-oriented format, and include them in our training corpus. These two sources together strengthen the model's reasoning and planning capabilities for complex decision-making tasks.
>
> **Q3:** For in-domain data curation, how are the questions and answers generated and how to ensure the quality and diversity of the dataset?
>
> **A3:** Thanks for the valuable suggestion. Due to space constraints, we initially described the generation process for different types of in-domain QA data in Appendix A.2 "In-Domain Data for Downstream VLAs" of the original submission, providing detailed prompting instructions and representative examples in Figures 3, 4, 5 in the Appendix. We use Qwen2.5VL-7B as the base model to generate the in-domain data items.
>
> To further enhance the quality and diversity of the generated in-domain QA data, as recommended by the reviewer, we additionally implement a
> data filtering pipeline as the post-processing process. Specifically, we adopt Qwen2.5VL-32B as judgement to score each generated data sample on
> a scale of 1 to 3 from the worst to best, while we filter out all samples assigned a score of 1, which account for approximately 10% of the initially generated data. The detailed data-filtering process and instruction prompts have also been added to the revised Appendix A.2 "In-Domain Data for downstream VLAs", and we use the refined in-domain data after filtering to train the newly-added VLA models presented in updated Table 2, Table 3 and Table 4.

---

> ### Author Response · Authors · 2025-11-24
> **To Reviewer ZtrG(Part 3)**
>
> **Q4:** The author need to incorporate an ablation over different chunk sizes.
>
> **A4:** Thanks for the valuable suggestion. We've added the ablation studies regarding chunk sizes in the revised Section 3.3 "Ablation studies", which is also presented below. We could observe that our proposed method still brings substantial improvements under different group of hyperparameter settings, which demonstrates great robustness of our method.
>
> | Model                                    | Predict Length | Execute Length | Sample Steps | Carrot on plate | Put eggplant in basket | Spoon on the towel | Stack cube | Avg    |
> | ---------------------------------------- | -------------- | -------------- | ------------ | --------------- | ---------------------- | ------------------ | ---------- | ------ |
> | **InternVL-2B** | 4              | 4              | 10           | 42.9%          | 57.1%                 | 55.8%             | 11.3%     | 41.8% |
> |                                          | 4              | 2              | 10           | 22.9%          | 18.3%                 | 40.8%             | 2.9%      | 21.2% |
> |                                          | 2              | 2              | 10           | 34.6%          | 22.9%                 | 54.2%             | 2.9%      | 28.7% |
> |                                          | 4              | 4              | 20           | 38.8%          | 54.2%                 | 51.3%             | 8.3%      | 38.2% |
> | **Vlaser-OOD(2B)** | 4              | 4              | 10           | 60.8%          | 35.4%                 | 56.7%             | 20.0%     | 43.2% |
> |                                          | 4              | 2              | 10           | 50.0%          | 21.7%                 | 30.0%             | 12.1%     | 28.5% |
> |                                          | 2              | 2              | 10           | 62.5%          | 19.2%                 | 49.2%             | 21.3%     | 38.1% |
> |                                          | 4              | 4              | 20           | 57.5%          | 29.2%                 | 54.6%             | 17.1%     | 39.6% |
> | **Vlaser-QA(2B)** | 4              | 4              | 10           | 55.8%          | 83.3%                 | 77.9%             | 33.3%     | 62.6% |
> |                                          | 4              | 2              | 10           | 44.2%          | 64.2%                 | 59.6%             | 36.3%     | 51.1% |
> |                                          | 2              | 2              | 10           | 47.5%          | 66.3%                 | 67.1%             | 36.3%     | 54.3% |
> |                                          | 4              | 4              | 20           | 56.3%          | 85.0%                 | 76.7%             | 35.0%     | 63.3% |

---

> ### Author Response · Authors · 2025-11-24
> **To Reviewer ZtrG(Part 4)**
>
> **Q5:** Please add the results for combining all in-domain data in Table 3 and clarify.
>
> **A5:** Thank you for this suggestion. In response, we have trained an additional **Vlaser-ALL** (2B) model, which is fine-tuned on the combination of all in-domain data types: general QA, spatial, and grounding. As shown in the updated Tables 2,3,4, incorporating any single type of in-domain data during pretraining leads to significant performance gains in VLA task execution, while combining all three data types yields further improvement. These results and corresponding analyses have been revised in Section 3.2 in the new updated manuscript. We also present the results for Google Robot (Table 3) below.
>
> | Model                 | Visual Matching   | Visual Matching | Visual Matching | Visual Matching | Variant Aggregation | Variant Aggregation | Variant Aggregation | Variant Aggregation |
> |----------------------|------------------------|--------------------|-----------------|--------|------------------------|--------------------|-----------------|--------|
> |                      | **Pick Coke Can** | **Move Near** | **Drawer** | **Avg** | **Pick Coke Can** | **Move Near** | **Drawer** | **Avg** |
> | InternVL3-2B         | 94.3%                | 78.8%             | 19.0%          | 64.0% | 80.4%                 | 72.7%             | 11.1%          | 54.7% |
> | Vlaser-OOD(2B)   | 85.0%                 | 76.3%             | 44.9%          | 68.7% | 74.4%                 | 69.2%             | 10.3%          | 51.3% |
> | Vlaser-QA (2B)  | 90.0%                 | 84.2%             | 44.4%          | 72.9% | 78.2%                 | 78.2%             | 13.0%          | 56.4% |
> | Vlaser-Spatial (2B) | 83.0%                 | 77.9%             | 56.0%          | 72.3% | 77.7%                 | 73.2%             | 13.2%          | 54.7% |
> | Vlaser-Grounding (2B) | 83.3%                 | 83.3%             | 54.2%          | 73.6% | 81.2%                 | 76.8%             | 17.0%          | 58.3% |
> | Vlaser-All (2B) | 91.0%                 | 85.4%             | 52.1%          | 76.2% | 80.5%                 | 77.7%             | 18.8%          | 59.0% |
>
> **Q6:** A further limitation is the lack of real-robot experiments that should at least be acknowledged in the limitations section.
>
> **A6:** Thank you for this insightful observation. During the rebuttal period, we have incorporated additional experiments using the RoboTwin dual-arm simulation platform, as now presented in Section 3.2 and Table 4 of the revised manuscript. RoboTwin has been widely recognized for its effectiveness and robustness in zero-shot sim-to-real transfer for bimanual manipulation tasks. We hope this could alleviate your concern on the effectiveness on real robots.
>
> Due to constraints on time and resources during the rebuttal period, we regard further validation on real-robots as an important direction for
> future work. We have explicitly acknowledged in the newly added Section 5 "Limitations" of our revision.
>
> [1] Andrew Szot, Max Schwarzer, Harsh Agrawal, Bogdan Mazoure, Rin Metcalf, Walter Talbott, Natalie Mackraz, R Devon Hjelm, and Alexander T Toshev. Large language models as generalizable policies for embodied tasks. In The Twelfth International Conference on Learning Representations, 2024.
>
> [2] Junhao Shi, Zhaoye Fei, Siyin Wang, Qipeng Guo, Jingjing Gong, and Xipeng Qiu. World-aware planning narratives enhance large vision-language model planner, arXiv preprint arXiv:2506.21230 (2025).

---

> ### Author Response · Authors · 2025-11-28
>
> Dear Reviewer ZtrG：
>
>    We sincerely appreciate your valuable time and effort in reviewing our manuscript and for providing the constructive feedback that helped improve our work. We have diligently incorporated substantial updates, including new experiments, detailed comparisons, and key clarifications to address all the points raised and strengthen the claims of our paper. As the rebuttal period is nearing its close, we would like to politely invite you to review our updated response and revised manuscript to ensure that we have fully and satisfactorily addressed all your concerns. If our rebuttal has successfully resolved your questions, **we would be deeply grateful for your reconsideration of the evaluation score to reflect the current state of the paper**. Should any residual concerns or questions remain, please do not hesitate to let us know, and **we will provide immediate further clarification**.
>
>    Thank you again for your patience and insight.
>
> Best regards, Authors

---

### Official Review · Reviewer_cVAr · 2025-11-01

**Soundness:** 3
**Presentation:** 3
**Contribution:** 3
**Rating:** 4
**Confidence:** 3

**Summary:**

This paper introduces Vlaser, a vision-language-action model that unifies embodied reasoning and robot control through a large-scale dataset and a dual-stage training pipeline. Built on InternVL3 and Qwen2.5 backbones with a flow-matching action expert, Vlaser achieves state-of-the-art results on 12 embodied reasoning benchmarks and strong performance on real-robot tasks like WidowX and Google Robot. The authors also provide a systematic analysis of data transfer, showing that in-domain embodied data is far more effective for policy learning than generic web data, highlighting a persistent domain gap between reasoning and control.

**Strengths:**

1. The paper presents a unified framework for embodied tasks and introduces the Vlaser-6M dataset.
2. The dataset is large, diverse, and carefully curated, covering grounding, spatial reasoning, planning, and simulation tasks.
3. The study offers useful observations on the gap between vision-language reasoning and low-level control, providing practical guidance for future model design.

**Weaknesses:**

1. The detailed architecture of the action expert is not presented. In Figure 2, the structure of the action expert is not shown.
2. I think the authors could provide more explanations or figures to help readers understand the structure of the action expert and the training process.
3. Since the paper emphasizes the impact of VLM reasoning on VLA performance, I think the authors could add some experiments on real robots or additional simulators to demonstrate the model’s generalization capability.

**Questions:**

1. The paper mentions “low-level control.” Does this refer to directly controlling the robot’s motors? This terminology seems to be commonly used in the robotics domain.
2. Although the paper describes the training and inference procedures, it is still unclear what exactly the pretrained VLM passes to the VLA.
3. In Figure 1, why are the closed-source models not included in the ranking?
4. Explain what is the open-loop inference and closed-loop control?
5. After adding the reasoning capability, does it affect the runtime efficiency of the VLA? Is this reasoning capability truly cost-effective?

---

> ### Author Response · Authors · 2025-11-24
> **To Reviewer cVAr(Part 1)**
>
> **Q1, Q2:** I think the authors could provide more explanations or figures to help readers understand the structure of the action expert and the training process.
>
> **A1, A2:** Thank you for your valuable feedback. Regarding the VLA end-to-end training and action expert structure, our approach follows prior flow matching based VLA methods [1,2], where a separate set of weights within the transformer layer is dedicated to processing robotics-specific (action and state) tokens, which are referred as *action expert*, resembling a mixture-of-experts (MoE) architecture. In response to your suggestion, we have updated Figure 2 in the revised manuscript to better illustrate the structure of the action expert (highlighted with yellow tokens). We have also expanded Section 2.3 "Vision-Language-Action Finetuning" with a more detailed description of the action expert design and the overall VLA training process, with more details and explanations for the flow-matching process. We believe these revisions provide a clearer explanation of the network architecture and the second-stage VLA fine-tuning procedure.
>
> **Q3:** I think the authors could add some experiments on real robots or additional simulators to demonstrate the model's generalization capability.
>
> **A3:** Thanks for the valuable suggestion. In response, we have incorporated additional experiments using the RoboTwin simulation platform - a well-established benchmark for dual-arm bimanual manipulation tasks. As presented in Section 3.2 and Table 4 of the revised manuscript (also included in the table below), we conducted closed-loop evaluations on 12 simulation tasks (from "Beat block hammer" to "Shake bottle") using the dual-arm Aloha-AgileX embodiment. The Vlaser model fine-tuned with in-domain data continues to achieve notable improvements over the InternVL3-2B and Vlaser OOD baselines, with an average task success rate increase of 5%-10%. We believe these additional experiments help demonstrate the generalizability of our method across different embodiments and tasks.
>
> | Simulation Task       | RDT-1B | InternVL3-2B | Vlaser-OOD (2B) | Vlaser-QA (2B) | Vlaser-Spatial (2B) | Vlaser-Grounding (2B) | Vlaser-All (2B) |
> | --------------------- | ------ | ------------ | --------------- | -------------- | ------------------- | --------------------- | --------------- |
> | Beat block hammer     | 28%    | 12%          | 20%            | 18%           | 20%                | 32%                  | 40%            |
> | Click bell            | 46%    | 78%          | 94%            | 48%           | 98%                | 86%                  | 92%            |
> | Handover mic          | 92%    | 74%          | 56%            | 84%           | 60%                | 52%                  | 84%            |
> | Move can pot          | 44%    | 40%          | 42%            | 66%           | 56%                | 50%                  | 46%            |
> | Move pillbottles pad  | 10%    | 62%          | 70%            | 66%           | 78%                | 68%                  | 72%            |
> | Move playingcard away | 20%    | 64%          | 52%            | 58%           | 68%                | 84%                  | 74%            |
> | Pick diverse bottles  | 2%     | 30%          | 44%            | 36%           | 24%                | 34%                  | 38%            |
> | Place burger fries    | 42%    | 36%          | 46%            | 88%           | 82%                | 46%                  | 42%            |
> | Place container plate | 82%    | 72%         | 70%            | 84%           | 78%                | 82%                  | 84%            |
> | Place phone stand     | 8%     | 42%         | 40%            | 38%           | 36%                | 48%                  | 50%            |
> | Place mouse pad       | 2%     | 68%         | 30%            | 44%           | 38%                | 48%                  | 92%            |
> | Shake bottle          | 66%    | 92%         | 90%            | 98%           | 96%                | 98%                  | 96%            |
> | Avg.                  | 36.8%  | 55.8%       | 54.5%          | 60.7%         | 61.2%              | 60.7%                | 67.5%          |

---

> ### Author Response · Authors · 2025-11-24
> **To Reviewer cVAr(Part 2)**
>
> **Q4:** What does "low-level control" refers to? Explain this terminology.
>
> **A4:** In our paper, "low-level control" denotes end-to-end policies that operate directly on the robot's joint states or end-effector pose, as opposed to higher-level multimodal perception and semantic understanding components.
>
> **Q5:** It is still unclear what exactly the pretrained VLM passes to the VLA.
>
> **A5:** As we figured out in Section 3.2, pretraining VLMs on out-of-domain embodied reasoning datas specific for embodied reasoning benchmarks brings litter improvements for VLA task execution (**Vlaser-OOD** vs **InternVL3-2B**). In contrast, pretraining with in-domain data including general QA, spatial, and grounding types leads to substantial performance gains. Based on the experiments, we reckon that pretraining VLMs on in-domain datas could significantly enhance the VLM's perception capability for the robotic in-domain ego-centric view images, thereby effectively reducing the visual observation domain shift when transferring from VLM pretraining to VLA fine-tuning.
>
> **Q6:** In Figure 1, why are the closed-source models not included in the ranking?
>
> **A6:** In the original Figure 1, we've already included closed-source model GPT-4o in the LiDAR visualization ranking. In response to the review's suggestion, we've also included another two closed-source models Claude-3.7-Sonnet and Gemini-2.5-Pro for ranking in the updated Figure 1.
>
> **Q7:** Explain what is the open-loop inference and closed-loop control.
>
> **A7:** Thank you for raising this point. We provide the following clarification. Open-loop inference refers to a single- or multi-turn question-answering process typically for VLMs, where the model's output does not influence subsequent inputs or the environmental state. This paradigm is often used to evaluate embodied reasoning capabilities, as reported in Table 1, where responses are generated based on a fixed context without real-time interaction. Closed-loop control, in contrast, describes the interactive task-execution process used by VLAs. Here, the action sequences generated by the VLA model directly affect the robot's next state and the resulting visual observations, which in turn form the input for the next decision step. This feedback loop continues until the task is completed or terminated, reflecting real-world deployment conditions. All VLA results in Tables 2-4 are based on this closed-loop evaluation setting.
>
> **Q8:** After adding the reasoning capability, does it affect the runtime efficiency of the VLA?
>
> **A8:** Thank you for raising this important question. We would like to clarify that our method follows a two-stage training paradigm, where embodied reasoning data is only used in the first-stage VLM pretraining to enhance VLM embodied reasoning capability. The second stage for VLA fine-tuning and the inference stage, following standard VLA protocols, directly predicts *N*-DOF action chunks without any additional thinking or reasoning process, thus would not affect the runtime efficiency. To reduce ambiguity, we have updated Figure 2 in the manuscript to better illustrate this two-stage design and the VLA model outputs.
>
> [1] Kevin Black, Noah Brown, Danny Driess, Adnan Esmail, Michael Equi, Chelsea Finn, Niccolo Fusai, Lachy Groom, Karol Hausman, Brian Ichter, et al. pi_{0}: A vision-language-action flow model for general robot control. arXiv preprint arXiv:2410.24164, 2024.
>
> [2] Physical Intelligence, Kevin Black, Noah Brown, James Darpinian, Karan Dhabalia, Danny Driess, Adnan Esmail, Michael Equi, Chelsea Finn, Niccolo Fusai, et al. pi_{0.5}: a vision-language-action model with open-world generalization. arXiv preprint arXiv:2504.16054, 2025.

---

> ### Author Response · Authors · 2025-11-28
>
> Dear Reviewer cVAr：
>
>    We sincerely appreciate your valuable time and effort in reviewing our manuscript and for providing the constructive feedback that helped improve our work. We have diligently incorporated substantial updates, including new experiments, detailed comparisons, and key clarifications to address all the points raised and strengthen the claims of our paper. As the rebuttal period is nearing its close, we would like to politely invite you to review our updated response and revised manuscript to ensure that we have fully and satisfactorily addressed all your concerns. If our rebuttal has successfully resolved your questions, **we would be deeply grateful for your reconsideration of the evaluation score to reflect the current state of the paper**. Should any residual concerns or questions remain, please do not hesitate to let us know, and **we will provide immediate further clarification**.
>
>    Thank you again for your patience and insight.
>
> Best regards, Authors

---

### Official Review · Reviewer_QnV2 · 2025-11-01

**Soundness:** 3
**Presentation:** 3
**Contribution:** 3
**Rating:** 6
**Confidence:** 3

**Summary:**

This paper presents Vlaser, a vision-language-action framework that couples a VLM for embodied reasoning with a flow-matching action expert for low-level control generation, supported by the large-scale Vlaser-6M data engine. The approach unifies grounding, spatial reasoning, and planning with closed-loop execution. Evaluations across 12 embodied benchmarks and SimplerEnv on WidowX and Google Robot demonstrate state-of-the-art performance and reveal that in-domain robot-view data drives the largest gains in downstream control tasks.

**Strengths:**

1. The paper is well-written, with clear figures and detailed explanations that make the work easy to follow.

2. The method introduces a clean end-to-end VLA design that decouples vision-language reasoning from an action expert and leverages flow matching for low-level control, enabling seamless closed-loop execution.

3. The paper demonstrates state-of-the-art performance across 12 embodied-reasoning benchmarks and consistently surpasses strong baselines.

4. It also conducts a systematic analysis of pretraining data effectiveness for VLA transfer and offers scaling insights.

**Weaknesses:**

1. The paper compares only two model scales within a single architecture and concludes that smaller models are better at point-based grounding while larger models excel at multi-step planning and closed-loop simulation; however, this scale–performance relationship may not generalize to other model families, and the work provides little deeper analysis or mechanistic explanation to substantiate it.

2. The policy relies on single-frame observations and fixed-length action sequences, with the action expert employing a fixed 10-step flow-matching sampler. This design limits the evaluation of temporal reasoning and multi-view robustness. The paper does not conduct ablations on these hyperparameters. The training uses an observation history of only one frame, which may constrain long-horizon planning capabilities.

**Questions:**

1. In Table 3, why does Vlaser exhibit lower performance than the InternVL base model? Could the authors include an ablation study that directly fine-tunes the base model on different in-domain datasets, i.e., to produce Vlaser-QA?

2. The evaluation covers only a small set of manipulation tasks on WidowX and Google Robot, making it difficult to claim generality to other embodiments or tasks. I recommend adding the second-part experiments and reporting the corresponding 8B results.

---

> ### Author Response · Authors · 2025-11-24
> **To Reviewer QnV2(Part 1)**
>
> **Q1:** The paper points that smaller models are better at point-based grounding while larger models excel at multi-step planning and closed-loop simulation, however, it may not generalize to other model families, and lacks deeper analysis.
>
> **A1:** Thank you for your valuable suggestion. We have indeed observed that for point-grounding capability, smaller models (e.g., 2B-3B) can perform comparably to or even better than larger ones (e.g., 7B-8B). Importantly, this trend appears to generalize beyond InternVL to other model families. For example, we also fine-tuned Qwen2.5-VL-3B and Qwen2.5-VL-7B on the same Vlaser-6M dataset and obtained the following results:
>
> |              | Embodied Grounding | Embodied Grounding | Embodied Grounding | Embodied Grounding | Planning & Simulation | Planning & Simulation | Planning & Simulation | Planning & Simulation |
> | ------------ | ------------------ | ------------------ | ------------------ | ------------------ | --------------------- | --------------------- | --------------------- | --------------------- |
> | **Model**    | **Where2place**    | **Pointarena**     | **Paco-Lavis**     | **Pixmo-Points**   | **Ego-Plan2**         | **VLABench**          | **EB-ALFRED**         | **EB-Habitat**        |
> | Qwen2.5VL-3B | 69.0               | 56.2               | 77.6               | 36.9               | 40.5                  | 31.2                  | 43.7                  | 31.1                  |
> | Qwen2.5VL-7B | 65.2               | 59.7               | 73.1               | 37.8               | 55.2                  | 44.4                  | 51.2                  | 38.9                  |
>
> A similar pattern has been reported in [RoBoBrain2.0](https://github.com/FlagOpen/RoboBrain2.0), which is based on the Qwen2.5-VL series. As shown in their Tables 2 to 4, RoBoBrain2.0-3B outperforms RoBoBrain2.0-7B on point-grounding benchmarks such as Where2Place and RefSpatial, while lagging behind on planning tasks like RoboBench Plan. Regarding the underlying principles, we hypothesize that point-grounding relies more heavily on perceptual capacity, which is largely associated with the vision transformer (ViT) in VLM architectures. In contrast, long-horizon planning demands stronger reasoning and thinking abilities, which are more tied to the LLM component. A plausible explanation is that both a 2B model and an 8B model often share the same vision encoder parameters (e.g., 300M InternViT), while differing mainly in the LLM component size (e.g., 1.78B vs. 7.61B in QwenLM).

---

> ### Author Response · Authors · 2025-11-24
> **To Reviewer QnV2(Part 2)**
>
> **Q2:** The paper lacks an ablation study and explanation of the hyperparameters used for the VLA, including frame number, action chunk size, and flow matching sampling steps.
>
> **A2:** Thank you for this valuable suggestion. We have now included ablation studies on the action chunk size and flow matching sampling steps in the table below. These results have also been added to Section 3.3 *Ablation Studies* in the revised manuscript. We could observe that our proposed method still brings substantial improvements under different group of hyperparameter settings, which demonstrates great robustness of our method.
>
> | Model                                    | Predict Length | Execute Length | Sample Steps | Carrot on plate | Put eggplant in basket | Spoon on the towel | Stack cube | Avg    |
> | ---------------------------------------- | -------------- | -------------- | ------------ | --------------- | ---------------------- | ------------------ | ---------- | ------ |
> | **InternVL-2B** | 4              | 4              | 10           | 42.9%          | 57.1%                 | 55.8%             | 11.3%     | 41.8% |
> |                                          | 4              | 2              | 10           | 22.9%          | 18.3%                 | 40.8%             | 2.9%      | 21.2% |
> |                                          | 2              | 2              | 10           | 34.6%          | 22.9%                 | 54.2%             | 2.9%      | 28.7% |
> |                                          | 4              | 4              | 20           | 38.8%          | 54.2%                 | 51.3%             | 8.3%      | 38.2% |
> | **Vlaser-OOD(2B)** | 4              | 4              | 10           | 60.8%          | 35.4%                 | 56.7%             | 20.0%     | 43.2% |
> |                                          | 4              | 2              | 10           | 50.0%          | 21.7%                 | 30.0%             | 12.1%     | 28.5% |
> |                                          | 2              | 2              | 10           | 62.5%          | 19.2%                 | 49.2%             | 21.3%     | 38.1% |
> |                                          | 4              | 4              | 20           | 57.5%          | 29.2%                 | 54.6%             | 17.1%     | 39.6% |
> | **Vlaser-QA(2B)** | 4              | 4              | 10           | 55.8%          | 83.3%                 | 77.9%             | 33.3%     | 62.6% |
> |                                          | 4              | 2              | 10           | 44.2%          | 64.2%                 | 59.6%             | 36.3%     | 51.1% |
> |                                          | 2              | 2              | 10           | 47.5%          | 66.3%                 | 67.1%             | 36.3%     | 54.3% |
> |                                          | 4              | 4              | 20           | 56.3%          | 85.0%                 | 76.7%             | 35.0%     | 63.3% |
>
> Regarding the frame number, we follow the typical VLA design [1,2,3] and use a single historical frame as input to ensure inference efficiency. The observation at each timestep $t$ is represented as $\mathbf{o}_t = [\mathbf{I}_t^1, ..., \mathbf{I}_t^n, l_t, \mathbf{q}_t]$ (image $\mathbf{I}_t^i$ with $n$ views, language $l_t$ and robot state $\mathbf{q}_t$). Note that the number of views $n$ varies with the embodiment (e.g., $n=1$ in Simpler and $n=3$ in RoboTwin), indicating that our approach readily supports multi-view inputs without compromising robustness. We hope these additions adequately address your concerns.

---

> ### Author Response · Authors · 2025-11-24
> **To Reviewer QnV2(Part 3)**
>
> **Q3:** In Table 3, why does Vlaser exhibit lower performance than the InternVL base model? Could the authors include an ablation study that directly fine-tunes the base model on different in-domain datasets?
>
> **A3:** Thank you for this valuable suggestion. Firstly, we would like to clarify that in the original Table 3, the entry "Vlaser(2B)" (last fourth from the bottom) with scores of 68.7% in visual matching and 51.3% in visual aggregation **is actually the Vlaser-2B model fine-tuned solely on Vlaser-6M out-of-domain (OOD) data specific for the embodied reasoning benchmarks in Sec. 3.1, without any in-domain data in Vlaser-6M dataset**. Its performance is indeed on par with the baseline model InternVL3-2B, which aligns with the statement in lines 412-413 of the original manuscript that *there is no positive correlation between common embodied reasoning benchmarks and the performance of closed-loop control of the lower level for the specific embodiment of the robot*. The performance improvements of our VLAs actually stem from fine-tuning the Vlaser-2B model on in-domain generated data-whether general QA, spatial QA, or grounding QA (as shown in the last three rows of Table 3). We apologize for the lack of clarity and any misunderstanding this may have caused. In the revised manuscript, we have updated the relevant parts of Section 3.2 and replaced "Vlaser" with "Vlaser-OOD" to clearly denote the out-of-domain model. We also present the results for Google Robot (Table 3) below. We hope these revisions resolve the confusion and better convey the contribution of our approach.
>
> | Model                 | Visual Matching   | Visual Matching | Visual Matching | Visual Matching | Variant Aggregation | Variant Aggregation | Variant Aggregation | Variant Aggregation |
> |----------------------|------------------------|--------------------|-----------------|--------|------------------------|--------------------|-----------------|--------|
> |                      | **Pick Coke Can** | **Move Near** | **Drawer** | **Avg** | **Pick Coke Can** | **Move Near** | **Drawer** | **Avg** |
> | InternVL3-2B         | 94.3%                | 78.8%             | 19.0%          | 64.0% | 80.4%                 | 72.7%             | 11.1%          | 54.7% |
> | Vlaser-OOD(2B)   | 85.0%                 | 76.3%             | 44.9%          | 68.7% | 74.4%                 | 69.2%             | 10.3%          | 51.3% |
> | Vlaser-QA (2B)  | 90.0%                 | 84.2%             | 44.4%          | 72.9% | 78.2%                 | 78.2%             | 13.0%          | 56.4% |
> | Vlaser-Spatial (2B) | 83.0%                 | 77.9%             | 56.0%          | 72.3% | 77.7%                 | 73.2%             | 13.2%          | 54.7% |
> | Vlaser-Grounding (2B) | 83.3%                 | 83.3%             | 54.2%          | 73.6% | 81.2%                 | 76.8%             | 17.0%          | 58.3% |
> | Vlaser-All (2B) | 91.0%                 | 85.4%             | 52.1%          | 76.2% | 80.5%                 | 77.7%             | 18.8%          | 59.0% |
>
> As recommended by the reviews, we have additionally include more separate results for Vlaser-2B model fine-tuned with different types of in-domain data - general QA, spatial, and grounding, referred to as **Vlaser-QA**, **Vlaser-Spatial**, and **Vlaser-Grounding**, respectively, along with the combined version **Vlaser-All**. These additions can be found in the revised Table 2, Table 3, and Table 4 of the updated manuscript (Note that Vlaser-QA, Vlaser-Spatial, and Vlaser-Grounding were already compared in the original Table 3 for Google Robot). We hope these expanded experiments will further help
> clarify the individual contributions of each in-domain data type and facilitate better reproducibility.

---

> ### Author Response · Authors · 2025-11-24
> **To Reviewer QnV2(Part 4)**
>
> **Q4:** The reviewer recommend adding the second-part experiments for other embodiments or tasks, and reporting the corresponding 8B results.
>
> **A4:** Thank you for your valuable suggestion. In response, we have incorporated additional experiments using the RoboTwin simulation platform - a well-established benchmark for dual-arm bimanual manipulation tasks. As presented in Section 3.2 and Table 4 of the revised manuscript (also included in the table below), we conducted closed-loop evaluations on 12 simulation tasks (from "Beat block hammer" to "Shake bottle") using the dual-arm Aloha-AgileX embodiment. The Vlaser model fine-tuned with in-domain data continues to achieve notable improvements over the InternVL3-2B and Vlaser-OOD baselines, with an average task success rate increase of 5%-10%. We believe these additional experiments help demonstrate the generalizability of our method across different embodiments and tasks.
>
> | Simulation Task       | RDT-1B | InternVL3-2B | Vlaser-OOD (2B) | Vlaser-QA (2B) | Vlaser-Spatial (2B) | Vlaser-Grounding (2B) | Vlaser-All (2B) |
> | --------------------- | ------ | ------------ | --------------- | -------------- | ------------------- | --------------------- | --------------- |
> | Beat block hammer     | 28%    | 12%          | 20%            | 18%           | 20%                | 32%                  | 40%            |
> | Click bell            | 46%    | 78%          | 94%            | 48%           | 98%                | 86%                  | 92%            |
> | Handover mic          | 92%    | 74%          | 56%            | 84%           | 60%                | 52%                  | 84%            |
> | Move can pot          | 44%    | 40%          | 42%            | 66%           | 56%                | 50%                  | 46%            |
> | Move pillbottles pad  | 10%    | 62%          | 70%            | 66%           | 78%                | 68%                  | 72%            |
> | Move playingcard away | 20%    | 64%          | 52%            | 58%           | 68%                | 84%                  | 74%            |
> | Pick diverse bottles  | 2%     | 30%          | 44%            | 36%           | 24%                | 34%                  | 38%            |
> | Place burger fries    | 42%    | 36%          | 46%            | 88%           | 82%                | 46%                  | 42%            |
> | Place container plate | 82%    | 72%         | 70%            | 84%           | 78%                | 82%                  | 84%            |
> | Place phone stand     | 8%     | 42%         | 40%            | 38%           | 36%                | 48%                  | 50%            |
> | Place mouse pad       | 2%     | 68%         | 30%            | 44%           | 38%                | 48%                  | 92%            |
> | Shake bottle          | 66%    | 92%         | 90%            | 98%           | 96%                | 98%                  | 96%            |
> | Avg.                  | 36.8%  | 55.8%       | 54.5%          | 60.7%         | 61.2%              | 60.7%                | 67.5%          |
>
> Regarding the 8B model results, we chose to maintain consistency with the model sizes commonly adopted in mainstream VLAs. Most popular VLA
> methods [2,3,4,5] typically use models under 4B parameters to ensure faster inference and efficient real-world deployment, thus we use Vlaser-2B as our primary VLA model. In this work, the Vlaser-8B model is primarily designed for high-level multimodal embodied reasoning capability, and we consider
> extending it to VLA settings as future work.
>
> [1] Moo Jin Kim, Karl Pertsch, Siddharth Karamcheti, Ted Xiao, Ashwin Balakrishna, Suraj Nair, Rafael Rafailov, Ethan Foster, Grace Lam, Pannag Sanketi, et al. Openvla: An open-source vision-language-action model. arXiv preprint arXiv:2406.09246, 2024.
>
> [2] Kevin Black, Noah Brown, Danny Driess, Adnan Esmail, Michael Equi, Chelsea Finn, Niccolo Fusai, Lachy Groom, Karol Hausman, Brian Ichter, et al. pi_{0}: A vision-language-action flow model for general robot control. arXiv preprint arXiv:2410.24164, 2024.
>
> [3] Physical Intelligence, Kevin Black, Noah Brown, James Darpinian, Karan Dhabalia, Danny Driess, Adnan Esmail, Michael Equi, Chelsea Finn, Niccolo Fusai, et al. pi_{0.5}: a vision-language-action model with open-world generalization. arXiv preprint arXiv:2504.16054, 2025.
>
> [4] Delin Qu, Haoming Song, Qizhi Chen, Yuanqi Yao, Xinyi Ye, Yan Ding, Zhigang Wang, JiaYuan Gu, Bin Zhao, Dong Wang, et al. Spatialvla: Exploring spatial representations for visual-language-action model. arXiv preprint arXiv:2501.15830, 2025.
>
> [5] Songming Liu, Lingxuan Wu, Bangguo Li, Hengkai Tan, Huayu Chen, Zhengyi Wang, Ke Xu, Hang Su, and Jun Zhu. Rdt-1b: a diffusion foundation model for bimanual manipulation. arXiv preprint arXiv:2410.07864, 2024

---

> ### Author Response · Authors · 2025-11-28
>
> Dear Reviewer QnV2：
>
>    We sincerely appreciate your valuable time and effort in reviewing our manuscript and for providing the constructive feedback that helped improve our work. We have diligently incorporated substantial updates, including new experiments, detailed comparisons, and key clarifications to address all the points raised and strengthen the claims of our paper. As the rebuttal period is nearing its close, we would like to politely invite you to review our updated response and revised manuscript to ensure that we have fully and satisfactorily addressed all your concerns. If our rebuttal has successfully resolved your questions, **we would be deeply grateful for your reconsideration of the evaluation score to reflect the current state of the paper**. Should any residual concerns or questions remain, please do not hesitate to let us know, and **we will provide immediate further clarification**.
>
>    Thank you again for your patience and insight.
>
> Best regards, Authors

---

### Author Response · Authors · 2025-11-24
**To All Reviewers and AC**

We sincerely thank all reviewers for their time, effort, and thoughtful evaluation of our paper, **as well as for their positive recognition of its contribution and novelty: We have conducted effective analysis of the transfer of pretraining vision-language data streams for VLA, revealing the gap between embodied reasoning and low-level control, and offering valuable insights into data transfer and scalability for embodied foundation models.** We have carefully considered all the comments and suggestions, and have incorporated all necessary revisions into the updated PDF file that has been submitted. These revisions will also be fully reflected in the final camera-ready version of the paper.

We thank the reviewers for their insightful feedback. The main concerns raised can be summarized as follows:

-   **More detailed ablation studies** are needed to quantify the individual contributions of each in-domain data source (noted by Reviewers QnV2, ZtrG, and HWMv).

-   **Additional experiments on other embodiments and tasks** are required to demonstrate the generalizability of the method (raised by Reviewers QnV2, cVAr, and ZtrG).

-   **Clarifications on the action-expert structure and flow-matching training process** are needed (requested by Reviewers cVAr and HWMv).

-   **Better explanation of the experimental results** in Table 3, particularly the entry "Vlaser (2B)" (mentioned by Reviewers QnV2 and HWMv).

In response to these valuable comments, we have made the following key revisions to the manuscript:

-   We have expanded Table 2, Table 3, and Table 4, and substantially revised Section 3.2, to include comprehensive ablation studies analyzing the individual contributions of each in-domain data type (Vlaser‑QA, Vlaser‑Spatial, Vlaser‑Grounding, and Vlaser‑All).

-   We have incorporated new experimental results on the RoboTwin simulation platform[1,2] for dual-arm bimanual manipulation tasks, presented in Table 4 and described in Section 3.2, to demonstrate broader applicability across embodiments.

-   We have updated Figure 2 for clearer illustration of the overall framework and action-expert architecture, and expanded Section 2.3 with a more detailed explanation of the flow-matching mechanism, including its mathematical formulation.

-   We have revised the notations and descriptions in Section 3.2, replacing "Vlaser (2B)" with "Vlaser‑OOD (2B)" to avoid ambiguity and improve interpretability.

We believe these revisions have thoroughly addressed the reviewers' concerns and significantly strengthened the contribution and clarity of the paper.

Furthermore, we commit to releasing the complete set of code, models, and data in an open-source community to facilitate reproducibility and future research. We deeply appreciate the reviewers' insightful and constructive feedback, and we kindly hope that you take our revisions and rebuttal into consideration for a possible reassessment and improvement of the paper's rating.

[1] Yao Mu, Tianxing Chen, Shijia Peng, Zanxin Chen, Zeyu Gao, Yude Zou, Lunkai Lin, Zhiqiang Xie, and Ping Luo. Robotwin: Dual-arm robot benchmark with generative digital twins (early version). In European Conference on Computer Vision, pp. 264–273. Springer, 2024

[2] Tianxing Chen, Zanxin Chen, Baijun Chen, Zijian Cai, Yibin Liu, Qiwei Liang, Zixuan Li, Xianliang Lin, Yiheng Ge, Zhenyu Gu, et al. Robotwin 2.0: A scalable data generator and benchmark with strong domain randomization for robust bimanual robotic manipulation. arXiv preprint arXiv:2506.18088, 2025c.

---

### Author Response · Authors · 2025-11-29
**The Summary Message to New AC**

Dear New Area Chair,

We want to first express our sincere gratitude for taking on the responsibility of handling our submission under these exceptionally challenging and unprecedented circumstances. Your commitment to maintaining ICLR’s high standards of quality and fairness in this unique case is highly appreciated.

We understand that you have been recently assigned this paper, and to assist your review, we would like to highlight the following critical points:

**1. Comprehensive Rebuttal and Manuscript Update**

As documented in our previous post titled **To All Reviewers and AC,** we have provided a comprehensive summary addressing all general reviewer concerns. This post details the **extensive new experiments** and **subsequent updates made to our manuscript** during the rebuttal period. We believe that these substantial revisions and added evidence have resolved all major concerns raised, and we sincerely hope you will give due consideration to the significant effort we put into refining this work during rebuttal period.

**2. Crucial Pre-Incident Score Update Timeline**

We wish to clarify the timeline of a critical score update that occurred *before* the massive reviewer identity leak incident (November 27th):

- **Positive Score Change:** Reviewer HWMv had already revised their score from a **4 to a 6** on **November 26th**, before the identity leak incident was revealed broadly. While this reviewer, regrettably, did not leave a corresponding text comment in the discussion log, we respectfully request that you verify this action by checking the **"Revisions"** history button within Reviewer HWMv’s specific review, **where the Rating has been modified to 6**. This pre-incident score change validates the positive recognition of our initial rebuttal and led us to post a **Thanks for raising the score** note on November 27th, prior to the incident.
- **Other Reviewers (QnV2, cVAr, ZtrG):** We regret that we were unable to receive further textual feedback from the remaining three reviewers before the reply window closed, despite our continuous follow-up. We appreciate their initial constructive comments and believe our comprehensive experimental additions have addressed their original concerns.

Finally, we respectfully draw your attention to the questions posed by this reviewer **cVAr** (Rating 4,Confidence 3). Based on some questions he raised  (Q1: Explain the terminology of ''low-level control'', Q4: Explain what is the open-loop inference and closed-loop control) which are foundational concepts in the embodied domain, we have some concern that the review may not be an domain-specific expert to fully appreciate and recognize the novelty and contribution of our work. While we believe we have addressed all his questions and concerns by incorporating the corresponding experiments and additional explanations in our revised manuscript (please see our rebuttal response), we kindly hope the new AC could apply their professional discretion regarding the weight assigned to this review during your final assessment.

Thank you again for your diligent service. We trust in your expertise to evaluate our submission fairly based on the totality of the above summary.

Sincerely,

Authors in Submission 4267

---

### Meta-Review · Area_Chair_AVLc · 2026-01-10

**Summary:**

This paper introduces Vlaser, a Vision-Language-Action (VLA) model designed to bridge high-level embodied reasoning with low-level robot control using a two-stage training approach and the large-scale Vlaser-6M dataset. Reviewers initially raised concerns regarding:


+ Architectural Clarity: The specific design of the action expert and the mathematical formulation of the flow-matching mechanism were insufficiently detailed.

+ Data Ablations: There was a lack of quantitative analysis regarding the individual contributions of different in-domain data types (QA, spatial, grounding) to downstream performance.

+ Evaluation Breadth: Initial results were limited to a few embodiments, and the claim regarding the lack of correlation between OOD reasoning benchmarks and control success required further evidence.

**Reviewer Concerns:**

**Addressed Concerns**

+ Technical and Mathematical Clarity: The authors updated Figure 2 and Section 2.3 to clarify the action-expert architecture and provided the specific mathematical formulation for the flow-matching mechanism.

+ Data Ablation Studies: New experiments were added to quantify the individual performance contributions of different in-domain data types (QA, spatial, and grounding).

+ Cross-Embodiment Generalization: The authors incorporated additional closed-loop evaluations on the RoboTwin simulation platform for dual-arm bimanual tasks.

+ Data Quality Assurance: A detailed LLM-as-a-judge filtering pipeline using Qwen2.5VL-32B was described to justify the quality and diversity of the Vlaser-6M dataset.

+ Terminology and Efficiency: Explanations for robotics-specific terminology and the runtime efficiency of the two-stage design were provided to resolve reviewer confusion.

**Outstanding Concerns**

+ Absence of Real-Robot Experiments: While the authors added more simulation benchmarks, they explicitly acknowledged that real-robot validation remains a limitation and a direction for future work.

+ Mechanistic Scaling Insights: While the authors provided additional empirical evidence for scaling behaviors across model families, the deeper mechanistic explanation for why model size affects specific capabilities differently remains largely hypothetical.

+ Temporal Reasoning Limits: The current model's reliance on a single-frame observation history may still pose long-term constraints for very complex, long-horizon tasks.

**Reviewer Scores:**

Reviewer HWMv: Officially revised their score from 4 to 6 on November 26th (pre-incident) after reviewing the initial rebuttal. Their concerns regarding mathematical clarity and data synergy were effectively addressed in the updated Section 2.3 and Figure 2.

Reviewer QnV2 (Current: 6): Likely to maintain a 6 or lean toward 7. The authors provided additional evidence from the Qwen2.5-VL family to substantiate their scaling claims and included the requested hyperparameter ablations.

Reviewer cVAr (Current: 4): Likely to increase to 5 or 6. The primary weaknesses—lack of architectural details, missing flow-matching math, and limited embodiment variety—were directly addressed through manuscript updates and the addition of the RoboTwin dual-arm benchmark.

Reviewer ZtrG (Current: 6): Likely to maintain a 6 or increase to 7. The authors provided the requested data-source ablations, verified the lack of evaluation set overlap, and added action-chunk-size ablations, satisfying most technical critiques.

---

### Decision · Program_Chairs · 2026-01-26

Accept (Poster)